# ADAPTING NOISE TO DATA:
# GENERATIVE FLOWS FROM LEARNED 1D PROCESSES

## ABSTRACT

We introduce a general framework for learning data-adaptive latent distributions (noise) in generative models based on 1D quantile functions through minimizing a statistical discrepancy between noise and data samples. Our quantile-based parameterization naturally adapts to heavy-tailed or compactly supported target distributions while shortening transport paths by capturing marginal structure. This construction, originally motivated by the study of 1D processes beyond the usual diffusion, integrates seamlessly with standard training objectives, including flow matching and consistency models. Numerical experiments highlight both the flexibility and the effectiveness of our approach, achieved with minimal computational overhead.

## 1 INTRODUCTION

Flow-based generative models, especially score-based diffusion Sohl-Dickstein et al. (2015); Song & Ermon (2019), flow matching (FM) Albergo et al. (2023); Lipman et al. (2023); Liu (2022) and one-step generative models (consistency models)Song et al. (2023); Boffi et al. (2025) like the recently introduced inductive moment matching (IMM) Zhou et al. (2025), achieve state-of-the-art results in many applications. All these methods construct a probability flow from a simple latent distribution (noise) to a complex target (data) with a neural network trained to approximate this flow from limited target samples. In diffusion models, the *score function* directs a reverse-time SDE, while in FM, the *velocity field* is learned to compute trajectories via a flow ODE. Finally, consistency models like IMM learn to predict the jumps from noise to the data while factoring in the consistency of the flow trajectories. Usually, a Gaussian is used as latent distribution which causes difficulties when learning certain multimodal and heavy-tailed targets, see Hagemann & Neumayer (2021); Salmona et al. (2022).A recent work of Ghane et al. (2025) has shown that diffusion models with Gaussian noise satisfy a concentration of measure property. Moreover, by Tam & Dunson (2025), GANs, VAEs and diffusion models with Gaussian or log-concave latent variables can only generate light-tailed samples and are not universal generators. See Figure 2 for a heavy-tailed example, where Gaussian noise fails. There exist only few approaches to learn the noising process, Bartosh et al. (2025) fit the forward diffusion process via a learned invertible map that is trained end-to-end, Kapusniak et al. (2024) use metric flow matching, i.e., a neural network to adapt the path to a underlying Riemannian metric. In a related approach Sahoo et al.

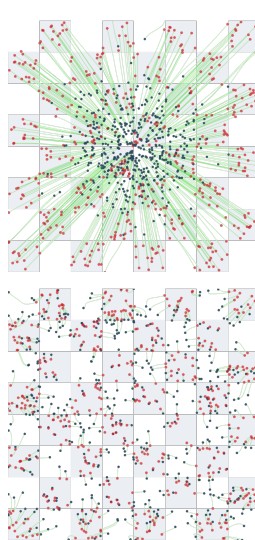

Figure 1: FM via optimal coupling with Gaussian noise (top) and our learned noise (bottom). Latent samples are shown in black, generated in red, and transportation paths in green. Starting from the learned latent drastically shortens the paths.

(2024) learns a input-conditioned componentwise Gaussian noise schedule. In the setting of sampling from unnormalized target densities, Blessing et al. (2025a) learn the latent noise by optimizing the mean and covariance of a Gaussian prior, while Blessing et al. (2025b) learn a Gaussian mixture prior, both are trained end to end. From a complementary perspective, Wiese et al. (2019) propose separating marginal modeling from dependence structure using copula and marginal flows, recognizing that standard architectures struggle with tail asymptotics, a motivation conceptually aligned with our componentwise quantile approach. On the other hand Pandey et al. (2024); Zhang et al. (2024) design heavy-tailed diffusions using Student-$t$ latent distributions, and Shariatian et al. (2025b) extend the framework to the family of $\alpha$-stable distributions.

In this paper, we present a new approach to adapt the latent distribution to the data by *learning* from its samples. The basic idea comes from the fact that all the above methods implicitly emerge as *componentwise* models. For example, denoting the target random variable by $\mathbf{X}_0$ and the latent by $\mathbf{X}_1 \sim \mathcal{N}(0, I_d)$, FM utilizes the process $\mathbf{X}_t = (X_t^1, \ldots, X_t^d)$ with the components $X_t^i = (1 - t)X_0^i + tX_1^i$ employing *one-dimensional* Gaussians $X_1^i \sim \mathcal{N}(0, 1)$. This motivated us to generally construct generative models from *1D processes and their quantile functions*.

Given any appropriate 1D process we demonstrate how to learn the componentwise neural flow by the associated conditional velocity field. We give examples besides diffusion demonstrating the flexibility of our machinery, namely the Kac process arising from the 1D damped wave equation, see Duong et al. (2025); Han et al. (2025), and a process reflecting the Wasserstein gradient flow of the maximum mean discrepancy with negative distance kernel towards the uniform distribution. In contrast to diffusion, assuming a compactly supported target, these processes also have a compact support, leading to a better regularity of the corresponding velocity field. This inspired us to further adapt the process to the data and to *learn* the 1D noising process rather than choosing it manually. To this end, we exploit that 1D probability measures can be equivalently described by their quantile functions $Q^i : (0, 1) \to \mathbb{R}$ which are monotone functions, and consider quantile processes $X_t^i = (1 - t)X_0^i + tQ^i(U^i)$, $i = 1, \ldots, d$ with i.i.d. $U^i \sim \mathcal{U}[0, 1]$ for $t \in [0, 1]$. We learn the individual quantile functions $Q_\phi^i$, $i = 1, \ldots, d$ such that their componentwise concatenation $\mathbf{Q}_\phi(\mathbf{U}) := (Q_\phi^i(U^i))_{i=1}^d$ is *"close"* to the data. This inspired us to minimize

$$W_2^2(\mu_0, \mathrm{Law}(\mathbf{Q}_\phi(\mathbf{U}))), \quad \mu_0 = \mathrm{Law}(\mathbf{X}_0).$$

with the Wasserstein distance $W_2$. We combine the learning of the latent $\mathbf{Q}_\phi(\mathbf{U})$ with the learning of the velocity field via optimal coupling FM. This allows us to effectively exploit the learned noise and drastically shorten the transport paths, as illustrated in Figure 1. The simplicity of quantile functions give us a flexible tool, which enables us to simultaneously learn the noising process and apply the FM framework. Our quantile perspective can further be extended to fit into consistency models.

**Contributions.** 1. We introduce a general construction method for generative neural flows by decomposing multi-dimensional flows into one-dimensional components. Ultimately, this allows us to work with *one-dimensional noising* processes in the FM framework.

2. Besides the usual Wiener process, we highlight two interesting noising processes: the physics-inspired 1D Kac process and the 1D MMD gradient flow with negative distance kernel leading to compactly supported noise and a better regularity of the FM velocity field.

3. The above hand-crafted processes motivate our *main contribution*: we propose to *learn the 1D noise distributions* themselves within the FM framework in a data adapted way, by parameterizing them through quantile functions and minimizing a statistical discrepancy. As a side result, our framework can also be incorporated into consistency models via so-called quantile interpolants.

4. Numerical experiments demonstrate that our method efficiently handles diverse marginal structures including heavy-tailed, compact, and multi-modal distributions. Learned quantiles shorten transport paths by capturing per-coordinate structure while delegating cross-dimensional dependencies to the velocity field.

## 2 FLOW MATCHING AND STOCHASTIC PROCESSES

We first review absolutely continuous curves in Wasserstein spaces as basis of the subsequent FM method. Then we highlight quite general stochastic processes $(X_t)_t$ "interpolating" between our target $X_0$ and a noising process $(Y_t)_t$ that starts in $Y_0 = 0$ and ends in $Y_1$ (our latent noise).

### 2.1 ABSOLUTELY CONTINUOUS CURVES IN WASSERSTEIN SPACE

We start with a brief introduction of curves in Wasserstein spaces and basic ideas on flow matching. For more details we refer to Ambrosio et al. (2008) and Wald & Steidl (2025). Let $(\mathcal{P}_2(\mathbb{R}^d), W_2)$ denote the complete metric space of probability measures with finite second moments equipped with the Wasserstein distance

$$W_2^2(\mu, \nu) := \min_{\pi \in \Pi(\mu, \nu)} \int_{\mathbb{R}^d \times \mathbb{R}^d} \|x - y\|^2 \, d\pi(x, y)$$

Here $\Pi(\mu, \nu)$ denotes the set of all probability measures on $\mathbb{R}^d \times \mathbb{R}^d$ having marginals $\mu$ and $\nu$. The push-forward measure of $\mu \in \mathcal{P}_2(\mathbb{R}^d)$ by a measurable map $\mathcal{T} : \mathbb{R}^d \to \mathbb{R}^d$ is defined by $\mathcal{T}_\sharp \mu := \mu \circ \mathcal{T}^{-1}$. Let $I$ be an interval in $\mathbb{R}$, in this paper mainly $I = [0, 1]$. A narrowly continuous curve $\mu_t : I \to \mathcal{P}_2(\mathbb{R}^d)$ is absolutely continuous, iff there exists a Borel measurable vector field $v : I \times \mathbb{R}^d \to \mathbb{R}^d$ with $\|v_t\|_{L_2(\mathbb{R}^d, \mu_t)} \in L_2(I)$ such that $(\mu_t, v_t)$ satisfies the continuity equation

$$\partial_t \mu_t + \nabla_x \cdot (\mu_t v_t) = 0 \tag{1}$$

in the sense of distributions. If in addition $\int_I \sup_{x \in B} \|v_t(x)\| + \text{Lip}(v_t, B) \, dt < \infty$ for all compact $B \subset \mathbb{R}^d$, then the ODE

$$\partial_t \varphi(t, x) = v_t(\varphi(t, x)), \qquad \varphi(0, x) = x, \tag{2}$$

has a solution $\varphi : I \times \mathbb{R}^d \to \mathbb{R}^d$ and $\mu_t = \varphi(t, \cdot)_\sharp \mu_0$.

Starting in the target distribution $\mu_0$ and ending in a simple latent distribution $\mu_1$, as usual in diffusion models, we can reverse the flow from the latent to the target distribution using just the opposite velocity field $-v_{1-t}$ in the ODE (2). Thus, if somebody provides us with the velocity field $v_t$, we can sample from a target distribution by starting in a sample from the latent one and then applying our favorite ODE solver.

### 2.2 FLOW MATCHING

If we do not have a velocity field donor, we can try to approximate (learn) the velocity field by a neural network $v_t^\theta$. Clearly, a desirable loss function would be

$$\mathcal{L}(\theta) := \mathbb{E}_{t \sim \mathcal{U}(0,1), \, x \sim \mu_t} \left[ \left\| v_t^\theta(x) - v_t(x) \right\|^2 \right].$$

Unfortunately this loss function is not helpful, since we do not know the exact velocity field $v_t$ nor can sample from $\mu_t$ in the empirical expectation. However, employing the law of total probabilities, as done, e.g. in Lipman et al. (2023), we see that $\mathcal{L}(\theta) = \mathcal{L}_{\text{CFM}}(\theta) + \text{const}$ with a constant not depending on $\theta$ and the conditional flow matching (CFM) loss

$$\mathcal{L}_{\text{CFM}}(\theta) := \mathbb{E}_{x_0 \sim \mu_0, \, t \sim \mathcal{U}(0,1), \, x \sim \mu_t(\cdot | x_0)} \left[ \left\| v_t^\theta(x) - v_t(x | x_0) \right\|^2 \right]. \tag{3}$$

The key difference is the use of the *conditional flow* $v_t(x | x_0)$ with respect to a fixed sample $x_0$ from our target distribution. To summarize, all you need is a *conditional* flow model with accessible velocity field $v_t(x | x_0)$ (at least along the flows trajectory), where you can easily sample from. Then you can indeed learn

the velocity field $v_t$ of the general (non-conditional) flow and finally sample from the target by the reverse ODE (2).

## 2.3 Stochastic Processes and Velocity Fields

Consider a continuously differentiable (noising) process $(\mathbf{Y}_t)_t$ with $\mathbf{Y}_0 \equiv 0 \in \mathbb{R}^d$ with associated velocity field $v_t = v_t^{\mathbf{Y}}(\cdot \mid 0)$ such that the pair $(\mu_t^{\mathbf{Y}}, v_t^{\mathbf{Y}})$ satisfy the continuity equation (1), where $\mu_t^{\mathbf{Y}}$ is the law of $(\mathbf{Y}_t)_t$[1]. To construct a generative model we need to create a process $(\mathbf{X}_t)_t$ which can start in any sample $x_0$ from the target measure $\mu_0$. Let $\mathbf{X_0} \sim \mu_0$. Following the lines in Duong et al. (2025), we define the *mean-reverting* process by

$$\mathbf{X}_t := f(t)\mathbf{X}_0 + \mathbf{Y}_{g(t)}, \quad t \in [0, 1], \tag{4}$$

with smooth *scheduling functions* $f, g$ fulfilling

$$f(0) = 1, \quad f(1) = 0 \quad \text{and} \quad g(0) = 0, \quad g(1) = 1. \tag{5}$$

Then we have $\mathbf{X}_1 = \mathbf{Y}_1$, and by abuse of notation, the process $\mathbf{X}_t$ starts in $\mathbf{X}_0 = \mathbf{X}_0$. Differentiation of (4) results in

$$\dot{\mathbf{X}}_t = \dot{f}(t)\mathbf{X}_0 + \dot{g}(t)\dot{\mathbf{Y}}_{g(t)}.$$

The conditional velocity field of $\mathbf{X}_t$ is given by (see Wald & Steidl (2025); Liu (2022))

$$v_t^{\mathbf{X}}(x \mid x_0) = \mathbb{E}\big[\dot{\mathbf{X}}_t \mid \mathbf{X}_t = x, \ \mathbf{X}_0 = x_0\big]$$

$$= \mathbb{E}\big[\dot{f}(t)\,x_0 + \dot{g}(t)\,\dot{\mathbf{Y}}_{g(t)} \ \big| \ \mathbf{Y}_{g(t)} = x - f(t)x_0\big]$$

$$= \dot{f}(t)\,x_0 + \dot{g}(t)\,v_{g(t)}^{\mathbf{Y}}\big(x - f(t)x_0 \mid 0\big). \tag{6}$$

Now, the conditional flow matching loss (3) can be minimized regarding $\mathbf{X}_0 \sim \mu_0$ and $\mathbf{X}_t \sim \mu_t$. Note that given a sample $x \sim (\mathbf{X}_t \mid \mathbf{X}_0 = x_0)$, we have $v_t^{\mathbf{X}}(x \mid x_0) = \dot{f}(t)\,x_0 + \dot{g}(t)\,v_{g(t)}^{\mathbf{Y}}\big(\mathbf{Y}_{g(t)} \mid 0\big)$. In general, $v^{\mathbf{Y}}$ might not be tractable, and only given as an conditional expectation of the time derivative $\dot{\mathbf{Y}}$. Yet, through our componentwise construction below, we will obtain easier access to it via its 1D components.

**Remark 1** (Relation to FM and diffusion). *Consider the stochastic process*

$$\mathbf{X}_t^{\mathrm{FM}} = \alpha_t\mathbf{X}_0 + \sigma_t\mathbf{X}_1, \qquad \mathbf{X}_1 \sim \mathcal{N}(0, I_d). \tag{7}$$

*Choosing $f(t) := \alpha_t$, $g(t) := \sigma_t^2$ and the standard Brownian motion $\mathbf{Y}_t = \mathbf{W}_t$, it holds the equality in distribution*

$$\mathbf{X}_t^{\mathrm{FM}} \overset{d}{=} f(t)\mathbf{X}_0 + \mathbf{W}_{g(t)} = \mathbf{X}_t.$$

*Then $f(t) := 1 - t$, $g(t) := t^2$ yields (independent) FM Lipman et al. (2023), and $f(t) := \exp\left(-\frac{h(t)}{2}\right)$, $g(t) := 1 - \exp\left(-h(t)\right)$, where $h(t) := \int_0^t \beta_{\min} + s(\beta_{\max} - \beta_{\min})\,\mathrm{d}s$ with, e.g., $\beta_{\min} = 0.1$, $\beta_{\max} = 20$, corresponds to processes used in score-based generative models Song et al. (2021), see Appendix B.*

**Remark 2** (Optimal Coupling). *Instead of considering (possibly independent) random variables $\mathbf{X}_0 \sim \mu_0, \mathbf{X}_1 \sim \mu_1$ and their induced processes (7), we can also employ optimal transport (OT) couplings $\pi \in \Pi_o(\mu_0, \mu_1)$. Then the induced curve $(e_t)_\sharp\pi$ with $e_t(x, y) := (1 - t)x + ty$ is a geodesic between $\mu_0$ and $\mu_1$ in $\mathcal{P}_2(\mathbb{R}^d)$. This yields an OT FM objective*

$$\mathcal{L}_{\mathsf{OT\text{-}CFM}}(\theta) = \mathbb{E}_{t\sim\mathcal{U}(0,1),\,(x,y)\sim\pi}\Big[\big\|v_\theta\big(e_t(x, y),\, t\big) - (y - x)\big\|_2^2\Big].$$

*In contrast to using the independent coupling, this can lead to reduced variance in training and both shorter and straighter paths, see Tong et al. (2024); Pooladian et al. (2023).*

Motivated by the fact that a multi-dimensional Wiener process $\mathbf{W}_t \in \mathbb{R}^d$ consists of *independent* (and identically distributed) 1D components $\mathbf{W}_t = (W_t^1, ..., W_t^d)$, we propose to construct a $d$-dimensional flow $\mathbf{Y}_t$ componentwise, based on independent one-dimensional processes $Y_t^i$.

---

[1]Existence of the velocity is given under weak assumptions by Wald & Steidl (2025) Theorem 6.3.

## 3   FROM ONE-DIMENSIONAL TO MULTI-DIMENSIONAL FLOWS

Restricting ourselves to processes $\mathbf{Y}_t$ that decompose into one-dimensional components allows us to propose a general construction method for accessible *conditional* flows in FM. Let $Y_t^1, \ldots, Y_t^d$ be a family of independent one-dimensional stochastic processes with time dependent laws $\mu_t^i \in \mathcal{P}_2(\mathbb{R})$. For each $i = 1, \ldots, d$, let $v_t^i \colon \mathbb{R} \to \mathbb{R}$ be the associated velocity field such that the pair $(\mu_t^i, v_t^i)$ satisfies the one-dimensional continuity equation (1). Define the product measure $\mu_t \in \mathcal{P}_2(\mathbb{R}^d)$ by

$$\mu_t(x) \;=\; \prod_{i=1}^d \mu_t^i(x^i), \qquad x = (x^1, \ldots, x^d) \in \mathbb{R}^d. \tag{8}$$

For the $d$-dimensional process $\mathbf{Y}_t := (Y_t^1, \ldots, Y_t^d)$, independence implies that its law is exactly $\mu_t$. Moreover, by the following proposition, the corresponding $d$-dimensional velocity field is given componentwise, see Holderrieth et al. (2025); Duong et al. (2025).

**Proposition 3.** *Let $\mu_t$ be given by* (8)*, where the $\mu_t^i$ are absolutely continuous curves in $\mathbb{R}$ with velocity fields $v_t^i$. Then $\mu_t$ satisfies a multi-dimensional continuity equation* (1) *with a velocity field which decomposes into the univariate velocities $v_t(x) := \left( v_t^1(x^1), \ldots, v_t^d(x^d) \right)$.*

Therefore, assuming access to the 1D velocities, we can **construct accessible conditional flows for FM**:

1. *One-dimensional noise:* Start with a 1D process and an associated absolutely continuous curve $\mu_t$ with $\mu_0 = \delta_0$, $0 \in \mathbb{R}$, where you can compute the velocity field $v_t$ in the 1D continuity equation

$$\partial_t \mu_t + \partial_x(\mu_t v_t) = 0, \qquad \mu_0 = \delta_0. \tag{9}$$

2. *Multi-dimensional noise:* Set up a multi-dimensional conditional flow model starting in $\mu_0 = \delta_0$, $0 \in \mathbb{R}^d$ with possibly different, but independent 1D processes as described in Section 3.

3. *Incorporating the data:* Construct a multi-dimensional conditional flow model starting in $\mu_0 = \delta_{x_0}$ for any data point $x_0 \sim \mu_0$ by mean-reversion as shown in Section 2.3.

To outline the use of this recipe, we explore three interesting 1D (noising) processes $Y_t$ in connection with their respective PDEs, for which our approach via reduction to one dimension is nicely applicable, namely the

- Wiener process $W_t$ and diffusion equation,

- Kac process $K_t$ and damped wave equation,

- Uniform process $U_t$ and the gradient flow of the maximum mean functional $\mathcal{F}_\nu := \mathrm{MMD}_K(\cdot, \nu)$ with negative distance kernel $K(x, y) = -|x - y|$ and $\nu = \mathcal{U}(-b, b)$.

In each case, we explicitly calculate the respective conditional measure flow and its conditional velocity field in Appendix A, such that the conditional flow matching loss (3) can be minimized. In each case, the absolutely continuous curve starting in $\delta_0$ and the corresponding velocity field can be calculated analytically. Note that in contrast to the Wiener process $W_t$ usually seen in diffusion and FM models, the latter two processes $K_t, U_t$ do not enjoy a trivial analogue in multiple dimensions: in case of $K_t$ the corresponding PDE (damped wave equation) is no longer mass-conserving in dimension $d \geq 3$, see Tautz & Lerche (2016); in case of $U_t$ the mere existence of the MMD gradient flow in multiple dimensions is unclear by the lack of convexity of the MMD, see Hertrich et al. (2024). Our general construction method makes these 1D processes accessible for generative modeling in arbitrary dimensions, hinting at a wide range of suitable noising processes.

---

[2]Note that we used the independent coupling for training of these models. We also used z-score normalization.

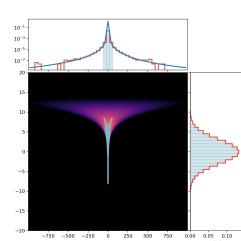 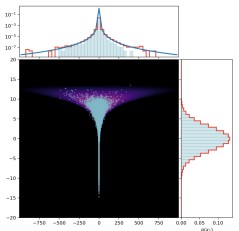 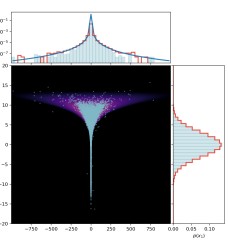 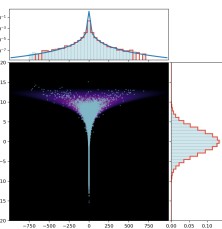

Figure 2: Sampling of Neal's funnel with different latent distributions.[2] Left to Right: uniform on $[-1, 1]$, standard Gaussian, Student-T (with parameters $(20, 4)$ inspired by the choice in Pandey et al. (2024)) and our learned distribution. The last two heavy-tailed noises perform significantly better.

## 4 ADAPTING NOISE TO DATA

Motivated by the influence of the noising process on the sample quality depicted for the heavy-tailed funnel in Figure 2, we propose to *learn* the noising process itself. We first revisit the connection between one-dimensional distributions and their quantile functions, then introduce quantile processes and quantile interpolants. Finally, we describe how the corresponding quantile functions can be learned in practice.

### 4.1 QUANTILE PROCESSES AND INTERPOLANTS

The restriction to componentwise noising processes $\mathbf{Y}_t$ in (4) [3] allows us to use the quantile functions of the 1D components. Recall that the *cumulative distribution function* (CDF) $R_\mu$ of $\mu \in \mathcal{P}_2(\mathbb{R})$ and its *quantile function* $Q_\mu$ are given by

$$R_\mu(x) := \mu\big((-\infty, x]\big), \quad x \in \mathbb{R} \quad \text{and} \quad Q_\mu(u) := \min\{x \in \mathbb{R} : R_\mu(x) \geq u\}, \quad u \in (0, 1). \quad (10)$$

In Figure 11, we exemplify the CDF and quantile function of a standard Gaussian. The quantile functions form a closed, convex cone $\mathcal{C} := \{f \in L_2(0, 1) : f \text{ increasing } a.e.\}$ in $L_2(0, 1)$. The mapping $\mu \mapsto Q_\mu$ is an isometric embedding of $(\mathcal{P}_2(\mathbb{R}), W_2)$ into $(L_2(0, 1), \|\cdot\|_{L_2})$, meaning that

$$W_2^2(\mu, \nu) = \int_0^1 \big|Q_\mu(s) - Q_\nu(s)\big|^2 \, \mathrm{d}s$$

and $\mu = Q_{\mu, \sharp} \mathcal{L}_{(0,1)}$. Let $U \sim \mathcal{U}[0, 1]$ be uniformly distributed on $[0, 1]$. Now, any probability measure flow $\mu_t$ can be described by their respective quantile flow $Q_t := Q_{\mu_t}$, such that $\mu_t = Q_{t, \sharp} \mathcal{L}_{(0,1)}$ and $Q_t \circ U$ is a stochastic process with marginals $\mu_t$.

**Quantile Processes.** We can therefore model any *multi-dimensional* noising process, that decomposes into its components, via quantile functions. Namely let $X_0$ be any component $\mathbf{X}_0^i$ of $\mathbf{X}_0 \sim \mu_0$, and $f, g : [0, 1] \to \mathbb{R}$ smooth schedules fulfilling (5). We assume that we are given a flow $(Q_t)_t$ of quantile functions $Q_t : (0, 1) \to \mathbb{R}$, $t \in [0, 1]$, which fulfill $Q_0 \equiv 0$ and are invertible on their respective image with the inverse given by the CDF $R_t : Q_t(0, 1) \to \mathbb{R}$. We introduce the *quantile process*

$$Z_t = f(t)X_0 + Q_{g(t)}(U), \quad U \sim \mathcal{U}(0, 1), \, t \in [0, 1]. \quad (11)$$

The quantile process coincides (in distribution) with the components of the mean-reverting process (4), where the noising term is represented as $\mathbf{Y}_{g(t)}^i \stackrel{d}{=} Q_{\mathrm{Law}(\mathbf{Y}_{g(t)}^i)}(U)$. In particular, the components of the process (7)

---

[3]Besides componentwise 1D processes we may also use triangular decompositions, not addressed in this paper.

are obtained via (11) using the quantile distribution $Q_t$ of a standard Brownian motion $W_t$ and $f(t) \coloneqq \alpha_t$, $g(t) \coloneqq \sigma_t^2$.

**Quantile Interpolants.** Let us briefly mention how our setting fits into the framework of consistency models. To this end, we define the *quantile interpolants*

$$I_{s,t}(x,y) = f(s)x + Q_{g(s)}\big(R_{g(t)}(y - f(t)x)\big), \quad s, t \in [0,1] \tag{12}$$

which generalize the interpolants used in Denoising Diffusion Implicit Models (DDIM), see Remark 10.

**Proposition 4.** *For all $x, y \in \mathbb{R}$ and all $s, r, t \in [0,1]$, it holds $I_{0,t}(x,y) = x$, $I_{t,t}(x,y) = y$, and*

$$I_{s,r}(x, I_{r,t}(x,y)) = I_{s,t}(x,y).$$

*Furthermore, inserting the quantile process* (11) *yields $I_{s,t}(Z_0, Z_t) = Z_s$.*

The proof is given Appendix C. Proposition 4 allows us to also apply the concept of consistency models to our quantile process (11). In the Appendix C, we demonstrate this by means of the recently proposed *inductive moment matching* (IMM) Zhou et al. (2025).

## 4.2 LEARNING QUANTILE FUNCTIONS

The choice of the noise can have a significant impact on the sampling performance, see Figure 1 for the checkerboard distribution and Figure 2 for a heavy-tailed one. Now we adopt the quantile process view from Section 4.1 to learn data–adapted noise. For simplicity we will only consider noising processes defined as a deterministic scaling of a *fixed* random variable $\mathbf{Z}$. We adopt a signal-decay schedule $f(t) = 1 - t$ and the linear latent growth $g(t) = t$, and consider $\mathbf{Y}_t \coloneqq t\mathbf{Z}$ and $v_t^{\mathbf{Y}}(x) = \frac{x}{t}$. Note that this corresponds to the standard linear interpolation often employed in FM. See Appendix A.4 for more theoretical background.

We pose the following requirements on the latent distribution $\nu$: i) data–independence, and ii) independence of components. Under these assumptions the latent class reduces to the set $S \coloneqq \{\nu \in \mathcal{P}_2(\mathbb{R}^d) : \nu = \rho \, dx \text{ and } \rho = \Pi_{i=1}^d \rho^i\}$, i.e. considering quantile processes of the form

$$X_t^i = (1-t)\, X_0^i \;+\; t Q^i(U^i), \; i = 1, \ldots, d, \; t \in [0,1],$$

we have $\nu = \mathbf{Q}_\# \, \mathcal{U}([0,1]^d)$ with $\mathbf{Q}(u) \coloneqq (Q^1(u^1), \ldots, Q^d(u^d))$. In particular, in our framework the quantile family determines the scales and tails of $\mathbf{Q}(\mathbf{U})$, thereby influencing the difficulty and inductive bias of predicting the conditional velocity $v_t(\mathbf{X}_t) = \mathbf{Q}(\mathbf{U}) - \mathbf{X}_0$ along the linear paths $\mathbf{X}_t = (1-t)\mathbf{X}_0 + t\mathbf{Q}(\mathbf{U})$.

We now describe how we learn the quantile maps $\mathbf{Q}_\phi$. The core idea is that besides our requirements i)-ii) as well as being a valid quantile function, we would like our noise to be *"close"* to the data. We learn $\mathbf{Q}_\phi$ by minimizing a statistical discrepancy, e.g. the Wasserstein distance, between $\mu_0$ and $\nu_\phi$,

$$\mathcal{L}_{\mathrm{AN}}(\phi) = W_2^2\big(\mu_0, \nu_\phi\big), \quad \nu_\phi \coloneqq (\mathbf{Q}_\phi)_\# \, \mathcal{U}([0,1]^d). \tag{13}$$

Note that due to the restriction of our quantiles to the class $S$, the minimizer of (13) is in general *not* $\mu_0$. Crucially, the independence constraint restricts $\nu_\phi$ to per-coordinate adaptation and prevents encoding *cross-dimensional* correlations. The latter are introduced via the optimal transport coupling $(x, y)$ and modeled by the velocity field through the target $(y - x)$. This separation lets the latent remain simple and computationally efficient while delegating dependencies to the flow.

While our quantiles can be trained independently, in order to provide an aligned training signal for the velocity field, we propose to also train $\mathbf{Q}_\phi$ *jointly* with the velocity $v_\theta$. Hence, we aim to minimize the loss

$$\mathcal{L}(\theta, \phi) = \mathcal{L}_{\mathsf{CFM}}(\theta, \phi) + \lambda \, \mathcal{L}_{\mathrm{AN}}(\phi), \quad \lambda > 0,$$

$$\text{with} \quad \mathcal{L}_{\mathsf{CFM}}(\theta, \phi) = \mathbb{E}_{t \sim \mathcal{U}(0,1),(x,y_\phi) \sim \pi_\phi}\Big[\big\| v_\theta\big((1-t)x + t y_\phi,\, t\big) - (\mathrm{sg}(y_\phi) - x)\big\|_2^2\Big],$$

where $\pi_\phi \in \Pi_o(\mu_0, \nu_\phi)$ is an optimal coupling between $\mu_0$ and $\nu_\phi$ and $\text{sg}(\cdot)$ denotes the stop-gradient operator . For reference we visualize the effect of choosing $\lambda = 0$ in Figure 15.

In practice, we optimize the empirical expectation via minibatches; for more details on the implementation see Appendix D.4. A pseudo-algorithm is provided in Algorithm 1. In particular, we compute a mini batch optimal transport map $T$ that minimizes $\sum_{j=1}^{B} \|\mathbf{x}_0^{(j)} - \mathbf{y}^{(T(j))}\|_2^2$ for batches of data $\{\mathbf{x}_0^{(j)}\}_{j=1}^{B}$, $\{\mathbf{y}^{(j)}\}_{j=1}^{B}$ from $\mathbf{X}_0$ and $\mathbf{Q}_\phi(\mathbf{U})$, respectively. This minibatch map $T$ is reused below for flow matching to keep the targets consistent across the two terms.

## 5 EXPERIMENTS

To provide intuition and validate our proposed method, we conduct experiments on both synthetic and image datasets. For each component, we model the quantile with a Rational Quadratic Spline (RQS) proposed in Gregory & Delbourgo (1982); Durkan et al. (2019) and add a learnable scale and bias. This keeps monotonicity, is parameter-efficient, and gives analytic derivatives. See Appendix D.2 for details.

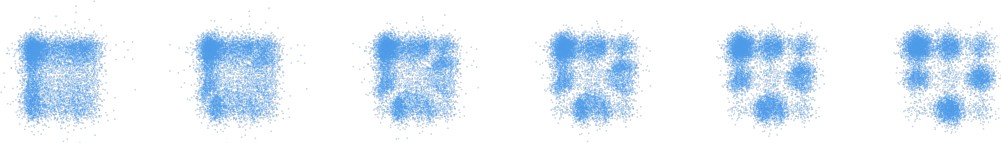

Figure 3: A generated trajectory from the learned quantile latent (left) to the unevenly weighted Gaussian mixture target (right). The adapted latent is already close to the target distribution.

### 5.1 SYNTHETIC DATASETS

We begin by qualitatively analyzing our algorithm on several synthetic 2D distributions, see also Appendix D.3, each designed to highlight a specific aspect of our approach. We provide intuition about the learned latent distribution and demonstrate that it is closer to the data in the Wasserstein sense, yields shorter transport paths, and successfully captures the tail behavior.

**Gaussian Mixture Model (GMM).** We first consider a 2D GMM with nine unevenly weighted modes, as visualized in Figure 3. Due to the independence assumption inherent in our factorized quantile function, the learned latent cannot perfectly replicate the target's joint distribution and is *not the product of the correct marginals*, see also Example D.1. Instead, it approximates a distribution where the components cannot further independently improve the transport cost to the target.

**Funnel Distribution.** The funnel distribution, shown in Figure 2, presents a challenge due to its heavy-tailed, conditional structure. several methods for handling it have been proposed in the context of diffusion models, e.g. Pandey et al. (2024); Shariatian et al. (2025a;b). We compare our method to Pandey et al. (2024), where the parameters of a latent Student-$t$ distribution were hand-select in each dimension. To visualize the effects more clearly, we use a capacity-constrained network with three layers of width 64 and no positional embeddings. This experiment highlights the importance of matching the latent's tail behavior to that of the target distribution, showing that a compact latent performs worst, followed by the Gaussian. At the same time, we observe that our learned latent successfully *adapts to the target's heavy tails*, see also Figure 13 in the appendix. This enables the FM model to generate high fidelity samples across the distribution. Note that due to the high variance signal when training on the funnel distribution, we pre-train our quantile.

**Checkerboard Distribution.** In contrast to the funnel, the checkerboard distribution in Figure 12 features a compact support. Here, we demonstrate the synergy between our learned latent and an OT coupling. Our method learns a latent that approximates a uniform distribution over the target's support. When this adapted latent is combined with an OT coupling for FM, the resulting *transport paths are substantially shorter* than those originating from a standard Gaussian as shown in Figure 1 in the appendix. Further, the vector field training converges much faster, see Figure 14 in the appendix. This result underscores our central claim: combining a data-dependent latent with a data-dependent coupling has the potential to significantly improve model performance.

## 5.2 IMAGE DATASETS

Next, we analyze our method on standard image generation benchmarks. Our quantile is extremely lightweight compared to the UNet architecture used for the flow model. We reuse the minibatch OT coupling for the latent and freeze the quantile function after a $55k$ training epochs. This strategy introduces only minimal computational overhead compared to the standard Gaussian baseline with minibatch OT coupling. On the CIFAR10 dataset for example, we observe an overhead of approximately $3.2\%$ in runtime during joint quantile training, and about $1.2\%$ after freezing the quantile parameters, measured on an NVIDIA GeForce RTX 4090. In high-dimensional settings and given fixed batch sizes, the signal for the quantile function can be noisy, potentially leading to degenerate solutions. To mitigate this, we add a regularization term to the loss that penalizes the expected negative log-determinant of the Jacobian of the quantile. Since the quantile maps from a uniform distribution $U \sim \mathcal{U}[0,1]$, this term equals the negative differential entropy of the learned latent

$$H(Q(U)) = H(U) + \mathbb{E}[\log|\det J_Q(U)|] = \mathbb{E}[\log|\det J_Q(U)|].$$

Access to analytic derivatives makes this efficient. For more details on the parametrization and its practicability see D.2.

**MNIST.** The MNIST dataset exhibits strong marginal structure: pixels near the center are frequently active (non-zero), whereas pixels at the borders are almost always zero. Our learned quantile function successfully captures these marginal statistics. As illustrated in Figure 5, the latent distribution learns to concentrate its mass in regions corresponding to active pixels.

In Figure 16 in the appendix, we compare the learned and empirical quantiles on the MNIST dataset at different pixel locations $(x, y)$. Where the pixel is essentially black, the learned quantile concentrates around that value, whereas in the center regions, where uncertainty is higher, the quantiles remain spread around zero (gray), accurately reflecting the data variability. In Figure 4, we compare the performance under different network capacity constraints by evaluating our learned latent against a Gaussian latent. Both latents are trained using mini-batch OT. We use the quantile loss weight $\lambda = 1$ (Eq. 4.2) and regularization parameter $\beta = 0.1$ (see Eq. D.2). As observed in Figure 5, the learned latent successfully minimizes the distance between noise and data by removing redundant information while the independence assumption prevents the model from capturing specific spatial correlations (e.g. the shape of a digit). This enables the network to use the available parameters more efficiently and achieve better results with the same parameter count.

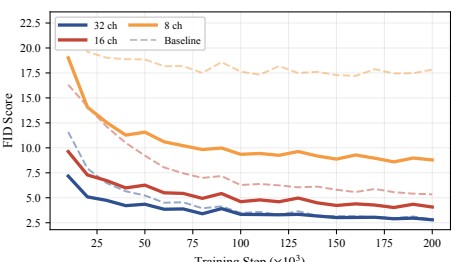

Figure 4: Ablation on U-Net capacity for MNIST using channels $8, 16, 32$. The FID curves show that our method achieves significantly lower FIDs when using less channels. See also Figure 17 the appendix.

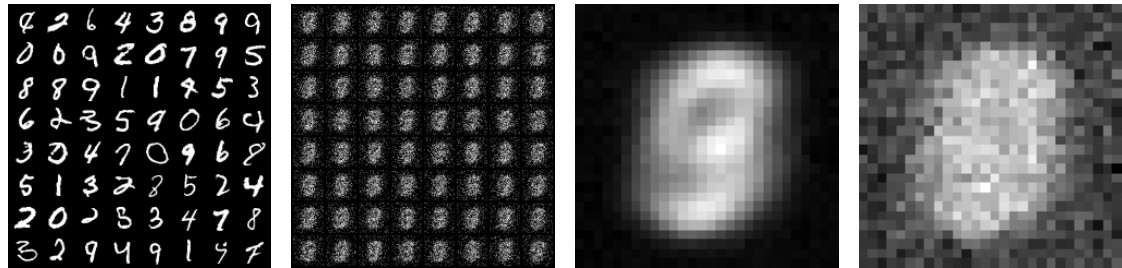

Figure 5: Left to Right: Generated samples, samples from the learned latent and mean and standard deviation of the learned latent.

**CIFAR-10.** On the CIFAR-10 dataset, we evaluate our method in a setting characterized by strong spatial and inter-channel correlations, where our product-measure approximations is inherently limited. We used a similar setup as in Tong et al. (2024) and including the commonly used U-Net architecture from Nichol & Dhariwal (2021). We vary the regularization parameter $\beta$ while keeping the quantile loss weight fixed at $\lambda = 1$. Figure 6 reports results for different values of $\beta$ and compares them to a standard Gaussian baseline. Our results indicate that for uncorrelated noise, there exists a trade-off between the entropy of the latent distribution and its closeness to the data. For independent noise on a highly correlated dataset, improvements remain marginal as expected since a product measure can only approximate the underlying data distribution to a limited extent. For more detail on training stability and the effect of regularization see Figures 19,20 in the appendix.

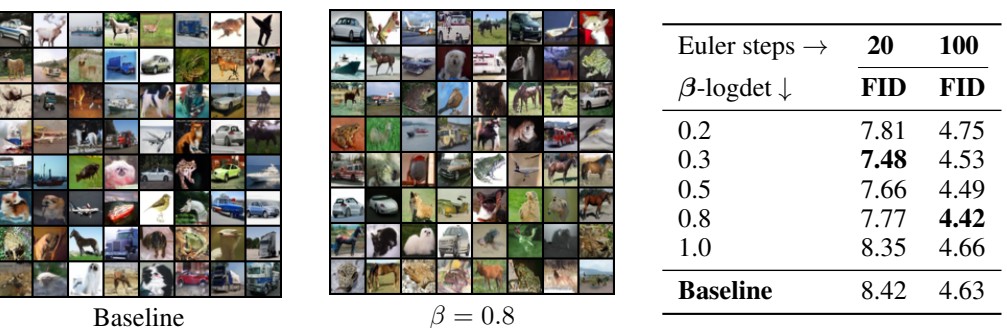

| Euler steps $\rightarrow$ | **20** | **100** |
|---|---|---|
| $\beta$-logdet $\downarrow$ | **FID** | **FID** |
| 0.2 | 7.81 | 4.75 |
| 0.3 | **7.48** | 4.53 |
| 0.5 | 7.66 | 4.49 |
| 0.8 | 7.77 | **4.42** |
| 1.0 | 8.35 | 4.66 |
| **Baseline** | 8.42 | 4.63 |

Baseline                     $\beta = 0.8$

Figure 6: CIFAR results for a selection of regularization parameters and for the baseline, for complete results see Figure 18. Our method reached the best validation FID after 320k steps, while the baseline took 340k. We evaluated the FID using 5 seeds and report the mean. We used those checkpoints for the evaluation. The visualized samples were generated using 100 Euler steps.

## 6 CONCLUSIONS

We provide a "quantile sandbox" for building generative models: a unifying theory and a practical toolkit that turns noise selection into a data-driven design element. Our construction plugs seamlessly into standard objectives including flow matching and consistency models, e.g. Inductive Moment Matching. Furthermore, our experiments demonstrate that it is possible to learn a freely parametrized, data-dependent latent distribution beyond the usual smooth transformations of Gaussians. Our work opens several promising directions for

future research. Extensions include developing time-dependent quantile functions to optimize the entire path distribution, not just the endpoint, as well as designing conditional quantile functions for tasks like class-conditional or text-to-image generation.

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

## A EXAMPLES OF ONE-DIMENSIONAL FLOWS

We provide three interesting examples, namely the well-established diffusion flow, the recently proposed Kac flow, and the Wasserstein gradient flow of the MMD functional with negative absolute distance kernel towards a uniform measure. Paths of the processes are depicted in Figure 7 and their probability flows are shown in Figures 8, 9 and 10.

In each case, the absolutely continuous curve $\mu_t$ starting in $\delta_0$ (e.g. conditional) and the corresponding velocity field can be given analytically. Note that in the latter two cases, multi-dimensional generalizations of the flows are not trivially given, which further underlines the strength of our 1D approach. Henceforth, if the measures $\mu_t$ admit a density function, we will denote it by $p_t$.

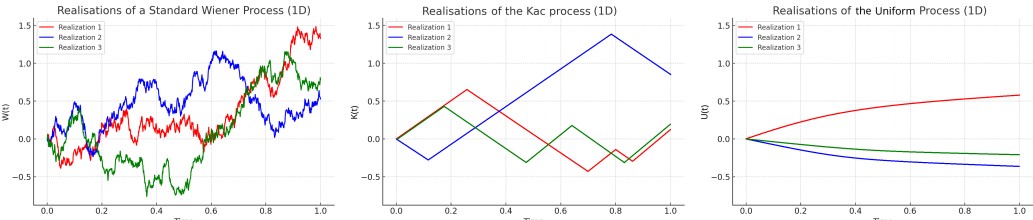

Figure 7: Three realisations of a standard Wiener process (left), the Kac process (middle), and the Uniform process (right), simulated until time $T = 1$.

### A.1 WIENER PROCESS AND DIFFUSION EQUATION

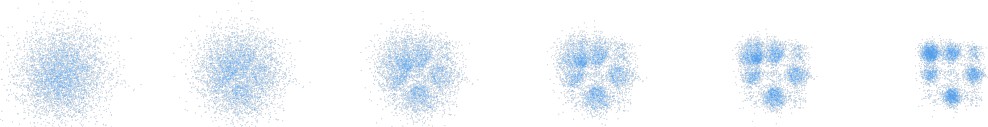

Figure 8: A generated trajectory from a Flow Matching model trained using the conditional density and velocity given by the Wiener process. As described in Section 2.3 we define the mean reverting process and use schedules $f(t) = 1 - t$ and $g(t) = t^2$.

First, consider the standard Wiener process (Brownian motion) $(W_t)_t$ starting in $0$ whose probability density flow $p_t$ is given by the solution of the diffusion equation

$$\partial_t p_t = \nabla \cdot (p_t \frac{1}{2} \nabla \log p_t) = \frac{1}{2} \Delta p_t, \quad t \in (0, 1], \qquad \lim_{t \downarrow 0} p_t = \delta_0, \tag{14}$$

where the limit for $t \downarrow 0$ is taken in the sense of distributions. The solution is analytically known to be

$$p_t(x) = (2\pi t)^{-\frac{d}{2}} e^{-\frac{\|x\|^2}{2t}}.$$

Thus, the latent distribution is just the Gaussian $p_1 = \mathcal{N}(0, I_d)$. The velocity field in (14) reads as

$$v_t(x) = -\frac{1}{2} \nabla \log p_t = \frac{x}{2t}. \tag{15}$$

However, its $L_2$-norm fulfills $\|v_t\|^2_{L_2(\mathbb{R},p_t)} = \frac{d}{4t}$, and is therefore not integrable over time, i.e. $\|v_t\|_{L_2(\mathbb{R},p_t)} \notin L_2(0,1)$. In practice, instability issues caused by this explosion at times close to the target need to be avoided by e.g. time truncations, see e.g. Kim et al. (2022). For a heuristic analysis also including drift-diffusion flows, we refer to Pidstrigach (2022). Note that in the case of diffusion, there is no significant distinction between the uni- and multivariate setting.

## A.2 KAC PROCESS AND DAMPED WAVE EQUATION

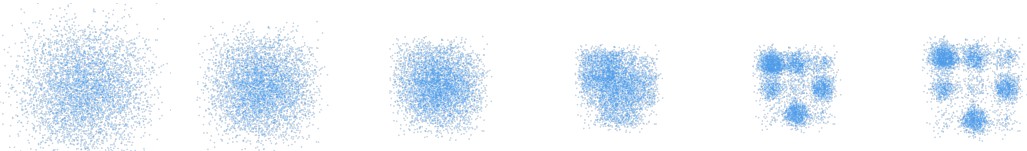

Figure 9: A generated trajectory from a flow matching model trained using the conditional density and velocity given by the Kac process with $(a,c) = (9,3)$. As described in Section 2.3 we define the mean reverting process and use schedules $f(t) = 1-t$ and $g(t) = t^2$.

The Kac process Kac (1974), also known as persistent random walk, originates from a discrete random walk, which starts in 0 and moves with velocity parameter $c > 0$ in one direction until it reverses its direction with probability $a\Delta_t$, $a > 0$. A continuous-time analogue is given by the Kac process which is defined using the homogeneous *Poisson point process* $N_t$ with rate $a$, i.e. i) $N_0 = 0$; ii) the increments of $N_t$ are independent, iii) $N_t - N_s \sim \text{Poi}\big(a(t-s)\big)$ for all $0 \le s < t$. Now the *Kac process starting in* 0 is given by

$$K_t := \text{B}_{\frac{1}{2}}\, c \int_0^t (-1)^{N_s}\, \mathrm{d}s, \tag{16}$$

where $\text{B}_{\frac{1}{2}} \sim \text{Ber}(\frac{1}{2})$ is a Bernoulli random variable[4] taking the values $\pm 1$. Note that in contrast to diffusion processes, the Kac process $K_t$ *persistently* maintains its linear motion between changes of directions (jumps of $N_t$), see Figure 7.

By the following proposition, the Kac process is related to the damped wave equation, also known as telegrapher's equation, and its probability distribution admits a computable vector field such that the continuity equation is fulfilled. For a proof we refer to Duong et al. (2025).

**Proposition 5.** *The probability distribution flow of $(K_t)_t$ admits a singular and absolutely continuous part via*

$$\mu_t(x) = \frac{1}{2}e^{-at}\big(\delta_0(x+ct) + \delta_0(x-ct)\big) + \tilde{p}_t(x), \tag{17}$$

*with the absolutely continuous part*

$$\tilde{p}_t(x) := \frac{1}{2}e^{-at}\Big(\beta ct\frac{I_0'(\beta r_t(x))}{r_t(x)} + \beta I_0(\beta r_t(x))\Big)1_{[-ct,ct]}(x), \qquad r_t(x) := \sqrt{c^2t^2 - x^2},$$

*where $\beta := \frac{a}{c}$, and $I_0$ denotes the 0-th modified Bessel function of first kind. The distribution (17) is the generalized solution of the damped wave equation*

$$\partial_{tt}u(t,x) + 2a\,\partial_t u(t,x) = c^2\partial_{xx}u(t,x), \tag{18}$$
$$u(0,x) = \delta_0(x), \qquad \partial_t u(0,x) = 0.$$

---

[4]More precisely, $\text{B}_{\frac{1}{2}}$ is *two-point* distributed with values $\{-1,1\}$.

*Further $(\mu_t, v_t)$ solves the continuity equation* (9) *where the velocity field is analytically given by*

$$
v_t(x) := \begin{cases}
\dfrac{x}{t + \frac{r_t(x)}{c} \frac{I_0(\beta r_t(x))}{I_0'(\beta r_t(x))}} & \text{if} \quad x \in (-ct, ct), \\
c & \text{if} \quad x = ct, \\
-c & \text{if} \quad x = -ct, \\
\text{arbitrary} & \text{otherwise}.
\end{cases}
$$

*The Kac velocity field admits the boundedness* $\|v_t\|_{L_2(\mathbb{R}, \mu_t)} \leq c$, *and hence,* $\|v_t\|_{L_2(\mathbb{R}, \mu_t)} \in L_2(0, 1)$.

Interestingly, the damped wave equation (18) is closely related to the diffusion equation via Kac' insertion method. It is based on the following theorem, whose proof based on semigroup theory can be found in Griego & Hersh (1971), see also Janssen (1990); Kac (1974).

**Theorem 6.** *For any initial function* $f_0 \in H^2(\mathbb{R}^d)$, $d \geq 1$, *let* $w_c(t, x)$ *be the solution of the* undamped *wave equation with velocity* $c > 0$ *given by*

$$
\partial_{tt} w(t, x) = c^2 \Delta w(t, x), \quad x \in \mathbb{R}^d, \ t > 0,
$$
$$
w(0, x) = f_0(x), \qquad \partial_t w(0, x) = 0.
$$

*Then, the functions defined by*

$$
h(t, x) := \mathbb{E}\left[w_1\left(\sigma W_t, x\right)\right], \quad \text{resp.} \quad u(t, x) := \mathbb{E}\left[w_c(c^{-1} S_t, x)\right]
$$

*solve the diffusion equation*

$$
\partial_t h(t, x) = \frac{\sigma^2}{2} \Delta h(t, x), \quad x \in \mathbb{R}^d, \ t > 0,
$$
$$
h(0, x) = f_0(x),
$$

*resp. the multi-dimensional damped wave equation*

$$
\partial_{tt} u(t, x) + 2a \, \partial_t u(t, x) = c^2 \Delta u(t, x), \quad x \in \mathbb{R}^d, \ t > 0,
$$
$$
u(0, x) = f_0(x), \qquad \partial_t u(0, x) = 0. \tag{19}
$$

As a consequence, it is not hard to show the following corollary, see Duong et al. (2025).

**Corollary 7.** *For any* $t \geq 0$, *the solution* $u^{a,c}(t, \cdot)$ *of the damped wave equation* (19) *converges to the solution* $h(t, \cdot)$ *of the diffusion equation for* $a, c \to \infty$ *with fixed* $\sigma^2 = \frac{c^2}{a}$.

In other words, diffusion can be seen as *"an infinitely $a$-damped wave with infinite propagation speed $c$"*. Note that the diffusion-related concept of particles traveling with infinite speed violates Einstein's laws of relativity and has therefore found resistance in the physics community Cattaneo (1958); Chester (1963); Vernotte (1958); Tautz & Lerche (2016).

We also like to stress that in multiple dimensions, the damped wave equation (18) is *no longer* mass-conserving as in 1D Tautz & Lerche (2016), and hence eludes a characterization via stochastic processes. Figure 9 shows the generation of samples from a weighted Gaussian Mixture Model (GMM) using Flow Matching and the Kac process as our noising process. As described in Section 2.3 we define the mean reverting process and use schedules $f(t) = 1 - t$ and $g(t) = t^2$.

### A.3 UNIFORM PROCESS AND MMD GRADIENT FLOW

Wasserstein gradient flows are special absolutely continuous measure flows whose velocity fields are negative Wasserstein (sub-)gradients of functionals $\mathcal{F}_\nu$ on $\mathcal{P}_2(\mathbb{R}^d)$ with the unique minimizer $\nu$. The gradient descent

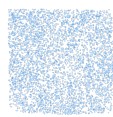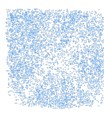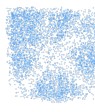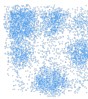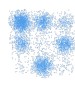

Figure 10: A generated trajectory from a flow matching model trained using the conditional density and velocity given by the MMD gradient flow. As described in Section 2.3 we define the mean reverting process and use schedules $f(t) = 1 - t$ and $g(t) = t$.

flow should reach this minimizer as $t \to \infty$. In this context, the MMD functional with the non-smooth negative distance kernel $K(x,y) = -|x - y|$ given by

$$\mathcal{F}_\nu(\mu) = \mathrm{MMD}^2_K(\mu, \nu) := -\frac{1}{2} \int_{\mathbb{R}^2} |x - y| \, \mathrm{d}\left(\mu(x) - \nu(x)\right) \, \mathrm{d}\left(\mu(y) - \nu(y)\right), \tag{20}$$

stands out for its flexible flow behavior between distributions of different support Hertrich et al. (2024). In 1D, its Wasserstein gradient flow $\mu_t$ can be equivalently described by the flow of its quantile functions $Q_{\mu_t}$ with respect to an associated functional on $L_2(0,1)$. Note that the MMD functional (20) loses its convexity (along generalized geodesics) in multiple dimensions Hertrich et al. (2024), and the general existence of their Wasserstein gradient flows is unclear in the multivariate case. This yields another reason to work in 1D, where we have have the following proposition.

**Proposition 8.** *The Wasserstein gradient flow $\mu_t$ of the MMD functional* (20) *starting in $\mu_0 = \delta_0$ towards the uniform distribution $\nu = \mathcal{U}[-b, b]$ with fixed $b > 0$ reads as*

$$\mu_t = \left(1 - \exp(-\tfrac{t}{b})\right) \mathcal{U}[-b, b], , \qquad t > 0, \tag{21}$$

*with corresponding velocity field*

$$v_t(x) = \frac{x}{b\left(\exp\left(\tfrac{t}{b}\right) - 1\right)}, \quad x \in \mathrm{supp}(\mu_t). \tag{22}$$

*It holds $\|v_t\|^2_{L_2(\mathbb{R}, \mu_t)} = \frac{2b}{3} \exp(-\frac{2t}{b})$, and hence, $\|v_t\|_{L_2(\mathbb{R}, \mu_t)} \in L_2(0,1)$. A corresponding (stochastic) process $(U_t)_t$ is given by $U_t := b\left(1 - \exp\left(-\frac{t}{b}\right)\right) U$, where $U \sim \mathcal{U}[-1, 1]$, such that $\mathrm{Law}(U_t) = \mu_t$.*

We prove the proposition more general for $\nu = \mathcal{U}[a, b]$ and a flow starting in $x_0 \in [a, b]$, i.e. we show

$$\mu_t = \mathcal{U}\left[a + (x_0 - a)\exp\left(-r(t)\right), b - (b - x_0)\exp\left(-r(t)\right)\right], \qquad t > 0 \tag{23}$$

with $r(t) := \frac{2t}{b-a}$ and

$$v_t(x) = \frac{2}{b - a}\left(\frac{x - x_0}{\exp(r(t)) - 1}\right). \tag{24}$$

To this end, we need the relation between measures in $\mathcal{P}_2(\mathbb{R})$ and cumulative distribution functions, see (10). For $\nu = \mathcal{U}[a, b]$, we have that

$$R_\nu(x) = \begin{cases} 0, & \text{if } x < a, \\ \frac{x-a}{b-a}, & \text{if } a \leq x \leq b, \\ 1, & \text{if } x > b \end{cases}$$

and $Q_\nu(s) = a(1-s) + bs$. In Hertrich et al. (2024) it was shown that the functional $F_\nu \colon L_2(0,1) \to \mathbb{R}$ defined by

$$F_\nu(u) := \int_0^1 \left( (1-2s)\big(u(s) + Q_\nu(s)\big) + \int_0^1 |u(s) - Q_\nu(t)| \, \mathrm{d}t \right) \mathrm{d}s \tag{25}$$

fulfills $\mathcal{F}_\nu(\mu) = F_\nu(Q_\mu)$ for all $\mu \in \mathcal{P}_2(\mathbb{R})$. Moreover, we have the following equivalent characterization of Wasserstein gradient flows of $\mathcal{F}_\nu$, which can be found in (Duong et al., 2024, Theorem 4.5).

**Theorem 9.** *Let $\mathcal{F}_\nu$ and $F_\nu$ be defined by* (20) *and* (25), *respectively. Then the Cauchy problem*

$$\begin{cases} \partial_t g(t) \in -\partial F_\nu(g(t)), & t \in (0, \infty), \\ g(0) = Q_{\mu_0}, \end{cases}$$

*has a unique strong solution $g$, and the associated curve $\gamma_t := (g(t))_\# \Lambda_{(0,1)}$ is the unique Wasserstein gradient flow of $\mathcal{F}_\nu$ with $\gamma(0+) = (Q_{\mu_0})_\# \Lambda_{(0,1)}$. More precisely, there exists a velocity field $v_t^*$ such that $(\gamma_t, v_t^*)$ satisfies the continuity equation* (9), *and it holds the relations*

$$v_t^* \circ g(t) \in -\partial F_\nu(g(t)) \quad \text{and} \quad v_t^* \in -\partial \mathcal{F}_\nu(\gamma_t). \tag{26}$$

Lastly note that here, the subdifferential $\partial F_\nu(u)$ is explicitly given by the singleton

$$-\partial F_\nu(u) = -\nabla F_\nu(u) = 2(\cdot - R_\nu \circ u) \quad \text{for all } u \in L_2(0,1),$$

see (Duong et al., 2024, Lemma 4.3).

*Proof of Proposition 8.* We want to apply Theorem 9 to $(\mu_t, v_t)$ in (23) and (24). The uniform distribution in (23) has the quantile function

$$Q_{\mu_t}(s) = \big(1 - \exp(-r(t))\big)\big(a + (b-a)s\big) + x_0 \exp(-r(t)), \qquad s \in (0,1).$$

For all $t > 0$ and all $s \in (0,1)$, we have $Q_{\mu_t}(s) \in [a,b]$ since $x_0 \in [a,b]$, and thus

$$-\nabla F_\nu(Q_{\mu_t})(s) = 2s - 2r_\nu(Q_{\mu_t}(s))$$

$$= 2s - 2\frac{\big(1 - \exp(-r(t))\big)\big(a + (b-a)s\big) + x_0 \exp(-r(t)) - a}{b-a}$$

$$= 2\left( s - \frac{x_0 - a}{b-a} \right) \exp(-r(t)).$$

On the other hand, it holds

$$\partial_t Q_{\mu_t}(s) = -2\frac{x_0 - a}{b-a} \exp(-r(t)) - \frac{(-2)(b-a)s}{b-a} \exp(-r(t)) = 2\left( s - \frac{x_0 - a}{b-a} \right) \exp(-r(t)).$$

By Theorem 9, $(\mu_t)$ is the unique Wasserstein gradient flow of $\mathcal{F}_\nu$ starting in $\delta_0$.

Furthermore, there exists a velocity field $v_t^*$ satisfying the continuity equation (9) and the relations (26). For $s \in (0,1)$ and $t > 0$, let $y := g_s(t) = a + (x_0 - a)\exp(-r(t)) + (b-a)(1 - \exp(-r(t)))s$. Then, we have $s = \frac{y - a - (x_0 - a)\exp(-r(t))}{(b-a)(1 - \exp(-r(t)))}$, and thus by (26),

$$v_t^*(y) = v_t^*(Q_{\mu_t}(s)) = 2\left( s - \frac{x_0 - a}{b-a} \right) \exp(-r(t))$$

$$= 2\left( \frac{y - a - (x_0 - a)\exp(-r(t))}{(b-a)(1 - \exp(-r(t)))} - \frac{x_0 - a}{b-a} \right) \exp(-r(t))$$

$$= \frac{2}{b-a}\left( \frac{y - a - (x_0 - a)}{1 - \exp(-r(t))} \right) \exp(-r(t))$$

$$= \frac{2}{b-a}\left( \frac{y - x_0}{\exp(r(t)) - 1} \right)$$

for all $y \in g_s(t)(0,1) = [a + (x_0 - a)\exp(-r(t)), b - (b - x_0)\exp(-r(t))]$. Lastly, let us compute the action. For $t > 0$ we have

$$\|v_t\|^2_{L^2(\mathbb{R},\mu_t)} = \int\limits_{a+(x_0-a)\exp\left(-\frac{2t}{b-a}\right)}^{b-(b-x_0)\exp\left(-\frac{2t}{b-a}\right)} \frac{4(x-x_0)^2}{(b-a)^2\left(\exp\left(\frac{2t}{b-a}\right)-1\right)^2} \frac{1}{(b-a)\left(1-\exp\left(-\frac{2t}{b-a}\right)\right)}\,\mathrm{d}x$$

$$= \frac{4}{(b-a)^3\left(\exp\left(\frac{2t}{b-a}\right)-1\right)^2\left(1-\exp\left(-\frac{2t}{b-a}\right)\right)} \int\limits_{a+(x_0-a)\exp\left(-\frac{2t}{b-a}\right)}^{b-(b-x_0)\exp\left(-\frac{2t}{b-a}\right)} (x-x_0)^2\,\mathrm{d}x$$

$$= \frac{4}{(b-a)^2\exp\left(-\frac{2t}{b-a}\right)\left(\exp\left(\frac{2t}{b-a}\right)-1\right)^3} \left[\frac{(x-x_0)^3}{3}\right]_{a+(x_0-a)\exp\left(-\frac{2t}{b-a}\right)}^{b-(b-x_0)\exp\left(-\frac{2t}{b-a}\right)}$$

$$= \frac{4\left(1-\exp\left(-\frac{2t}{b-a}\right)\right)^3}{3(b-a)^2\exp\left(-\frac{2t}{b-a}\right)\left(\exp\left(\frac{2t}{b-a}\right)-1\right)^3} \left[(b-x_0)^3 - (a-x_0)^3\right]$$

$$= \frac{4\left[(b-x_0)^3 - (a-x_0)^3\right]}{3(b-a)^2}\exp\left(-\frac{4t}{b-a}\right).$$

and the proof is finished. $\qquad\square$

Note that the fact that $v_t^*$ is uniquely determined on $\operatorname{supp}\mu_t = \overline{g_t(0,1)}$, correlates with the fact that the gradient $v_t^* \circ g(t) = -\nabla F_\nu(g(t))$ is a *singleton*. Outside of $\operatorname{supp}\mu_t$, the velocity field may be arbitrarily extended, which yields a velocity $\tilde{v}_t \in -\partial\mathcal{F}_\nu(\mu_t)$ in a *non-singleton* subdifferential. The velocity $v_t^*$ may be *uniquely* chosen from the tangent space $T_{\mu_t}\mathcal{P}_2(\mathbb{R})$, or equivalently, by choosing it to have minimal norm, i.e. $v_t^* \equiv 0$ outside of $\operatorname{supp}\mu_t$.

## A.4 SCALED LATENT DISTRIBUTIONS

Finally, we consider a simple class of processes obtained by a deterministic scaling of a latent random variable. In particular, we will see that the above Wiener process and the Uniform process are of this form, while the Kac process is not. Let $Z$ be a random variable with law $\rho_Z \in \mathcal{P}_2(\mathbb{R})$, and let $g\colon [0,1] \to [0,\infty)$ be continuously differentiable with $g(0) = 0$ and $g(1) = 1$. We consider

$$Y_t := g(t)\,Z, \qquad t \in [0,1],$$

with $Y_t \sim \mu_t$. Supposing that $\mu_t$ has density $\rho_t$, we get

$$\rho_t(x) = g(t)^{-d}\rho_Z\left(\frac{x}{g(t)}\right), \qquad t > 0, \quad \text{and} \quad \lim_{t\downarrow 0}\mu_t = \delta_0.$$

Then straightforward computation yields that $\mu_t$ together with the velocity field

$$v_t(x) = \frac{g'(t)}{g(t)}\,x, \qquad x \in \operatorname{supp}(\mu_t)$$

with the convention $v_t(0) = 0$ and arbitrary outside $\operatorname{supp}(\mu_t)$, solves the continuity equation (9). Further, it holds

$$\int_0^1 \|v_t\|^2_{L_2(\mathbb{R},\mu_t)}\,\mathrm{d}t = \mathbb{E}[\|Z\|^2]\int_0^1 \left(g'(t)\right)^2\mathrm{d}t < \infty \quad \text{whenever } g' \in L_2(0,1).$$

Also note that if $Z = Q(U)$ for a quantile function $Q : (0,1) \to \mathbb{R}$ and a random variable $U \sim \mathcal{U}([0,1])$, we have

$$\mathbb{E}[\|Z\|^2] = \int_0^1 |Q(u)|^2 \, du,$$

i.e. the second moment of $Z$ is exactly given by the $L_2$-norm of its quantile. Hence, explosions of the velocity's norm $\int_0^1 \|v_t\|^2_{L_2(\mathbb{R}, \mu_t)} \, dt$ can be directly controlled by the derivative of the time schedule $g$, and the size of the quantile function $Q$ of the latent variable.

The Wiener process fits into this framework with $g(t) = \sqrt{t}$ and $Z \sim \mathcal{N}(0,1)$, which recovers the exploding vector field $v_t(x) = \frac{1}{2t} x$ in (15). Also the Uniform process appears as a special case of the scaling process. In contrast, the Kac process does *not* belong to this class, as it is not generated by a deterministic scaling map but by persistent velocity switching, cf. (16).

# B  FLOW MATCHING AS SPECIAL MEAN REVERTING PROCESSES

## B.1  THE GAUSSIAN CASE

Let us shortly verify that our componentwise approach using the mean-reverting process (4), i.e.

$$\mathbf{X}_t := f(t)\,\mathbf{X_0} + \mathbf{Y}_{g(t)},$$

leads to the usual flow matching objective. where we choose the scheduling functions $f(t) := 1 - t$, $g(t) := t^2$, the target random variable $\mathbf{X}_0 \sim \mu_0$, and a standard Wiener process $\mathbf{Y}_t$ in $\mathbb{R}^d$ (independent of $\mathbf{X}_0$): First, it holds $\mathbf{Y}_{t^2} \sim \mathcal{N}(0, t^2 I_d)$, hence $\mathbf{Y}_{t^2} \stackrel{d}{=} t\,\mathbf{Z}$ with $\mathbf{Z} \sim \mathcal{N}(0, I_d)$, so that

$$\mathbf{X}_t \stackrel{d}{=} (1-t)\mathbf{X_0} + t\,\mathbf{Z}.$$

Furthermore, by (15) the 1D components of $\mathbf{Y}_t$ admit the velocity field $v_t^i(x^i) = \frac{x^i}{2t}$, $x^i \in \mathbb{R}$, and by Proposition 3 the multi-dimensional process $\mathbf{Y}_t$ admits the velocity field $v_{\mathbf{Y}}(t,x) = (\frac{x^1}{2t}, ..., \frac{x^d}{2t}) = \frac{x}{2t}$, $x = (x^1, ..., x^d) \in \mathbb{R}^d$. By the calculation (6), the conditional velocity field corresponding to $\mathbf{X}_t$ starting in $x_0 \in \mathbb{R}^d$ reads as

$$\begin{aligned}
v_{\mathbf{X}}(t, x \mid x_0) &= \dot{f}(t)\,x_0 + \dot{g}(t)\,v_{\mathbf{Y}}\big(g(t),\, x - f(t)x_0 \mid 0\big) \\
&= -x_0 + 2t\,v_{\mathbf{Y}}\big(t^2,\, x - (1-t)x_0 \mid 0\big) \\
&= -x_0 + \frac{x - (1-t)x_0}{t}.
\end{aligned}$$

Now, if $x \sim P_{\mathbf{X}_t}(\cdot \mid x_0)$, i.e. $x = (1-t)x_0 + tz$ with $z \sim \mathcal{N}(0, I_d)$, then it follows

$$v_{\mathbf{X}}(t, x \mid x_0) = -x_0 + \frac{(1-t)x_0 + tz - (1-t)x_0}{t} = z - x_0, \tag{27}$$

which is the usual constant-in-time conditional FM velocity along the straight-line trajectories between $x_0 \sim \mu_0$ and $z \sim \mathcal{N}(0, I_d)$.

## B.2  THE UNIFORM CASE

Now consider any component of the mean-reverting process (4) with $f(t), g(t)$ to be chosen, $X_0$ being a component of $\mathbf{X}_0 \sim \mu_0$, and $Y_t$ given by the MMD gradient flow (21), i.e. $Y_t := b\left(1 - \exp\left(-\frac{t}{b}\right)\right) U$, where $U \sim \mathcal{U}[-1, 1]$. Let $v_Y$ be the corresponding velocity field from (22). Then, we have

$$v_X(t, x | x_0) = \dot{f}(t)\, x_0 \;+\; \dot{g}(t)\, v_Y\big(g(t),\, |x - f(t)x_0|\big) \frac{x - f(t)x_0}{|x - f(t)x_0|}$$

$$= \dot{f}(t)\, x_0 \;+\; \dot{g}(t)\, \frac{x - f(t)x_0}{b\left(\exp\left(\frac{g(t)}{b}\right) - 1\right)}.$$

Now, along the trajectory $x \sim P_{X_t}(\cdot \mid x_0)$, i.e.

$$x \;=\; f(t)\, x_0 + b\left(1 - \exp\left(-\frac{g(t)}{b}\right)\right) u \;=:\; \alpha_t\, x_0 + \sigma_t\, u, \tag{28}$$

with $u \sim \mathcal{U}(-1, 1)$, the velocity calculates as

$$v_X(t, x \mid x_0) = \dot{f}(t)\, x_0 \;+\; \dot{g}(t)\, \frac{b\left(1 - \exp\left(-\frac{g(t)}{b}\right)\right) u}{b\left(\exp\left(\frac{g(t)}{b}\right) - 1\right)}$$

$$= \dot{f}(t)\, x_0 \;+\; \dot{g}(t)\, \exp\left(\frac{-g(t)}{b}\right) u$$

$$= \dot{\alpha}_t\, x_0 + \dot{\sigma}_t\, u, \tag{29}$$

where $\alpha_t := f(t)$ and $\sigma_t := b\left(1 - \exp\left(-\frac{g(t)}{b}\right)\right)$. Hence, in order to minimize the CFM loss, we only need to sample $t \sim \mathcal{U}[0, 1]$, $x_0 \sim X_0$, and $u \sim \mathcal{U}(-1, 1)$. Note the similarity between the MMD path (28) and the FM/diffusion path (7); by choosing $b = 1$, $f(t) := 1 - t$ and $g(t) := -\log(1 - t)$ it follows $\alpha(t) = 1 - t$, $\sigma(t) = t$, and we obtain in (29) the FM-velocity along the trajectory (27), where the Gaussian noise $z \sim \mathcal{N}(0, 1)$ is just replaced by a uniform noise $u \sim \mathcal{U}(-1, 1)$.

## C  IMM WITH QUANTILE INTERPOLANTS

In this section, we want to demonstrate how the IMM framework proposed in Zhou et al. (2025) can be realized by our quantile approach.

The general idea of consistency models is to predict the jumps from a process $Z_t$ to the target $X_0$, while factoring in the *consistency* of the trajectory of $Z_t$ via $Z_s$, $0 < s < t$. In FM, this consistency of the flow is usually neglected as only single points on the FM paths are sampled. Also, consistency models as one-step or multistep samplers usually are in no need of velocity fields.

Note that in the following – for notational simplicity – we consider the one-dimensional case $X_0, Z_t \in \mathbb{R}$ where we can employ quantile functions. By combining the 1D components into a multivariate model $\mathbf{X}_0 = (X_0^1, ..., X_0^d)$, $\mathbf{Z}_t = (Z_t^1, ..., Z_t^d)$, the results of this chapter trivially extend to $\mathbb{R}^d$.

Recall our definition of the *quantile process*

$$Z_t = f(t)X_0 + Q_{g(t)}(U), \quad U \sim \mathcal{U}(0, 1),\, t \in [0, 1]. \tag{30}$$

and the *quantile interpolants*

$$I_{s,t}(x, y) = f(s)x + Q_{g(s)}\big(R_{g(t)}(y - f(t)x)\big), \quad s, t \in [0, 1]. \tag{31}$$

Note that by the assumptions (5) it holds $Z_0 = X_0$ and $Z_1 = Q_1(U)$.

By the following remark, our quantile interpolants generalize the interpolants used in Denoising Diffusion Implicit Models (DDIM).

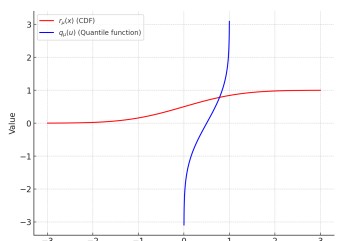

Figure 11: The CDF $R_\mu$ and quantile function $Q_\mu$ of a standard normal distribution $\mu$.

**Remark 10** (Relation to DDIM). *The interpolants used in Denoising Diffusion Implicit Models (DDIMs) Song et al. (2020) are given by*

$$\text{DDIM}_{s,t}(x,y) := \left(\alpha_s - \frac{\sigma_s}{\sigma_t}\alpha_t\right)x + \frac{\sigma_s}{\sigma_t}y. \tag{32}$$

*Now let $f(t) := 1 - t$, $g(t) := t^2$ and let $Q_t$ be the quantile of the law of a standard Brownian motion $W_t$.*

*First we obtain*

$$Q_{g(t)}(p) = Q_{t^2}(p) = Q_{\mathcal{N}(0,t^2)}(p) = t\sqrt{2}\,\text{erf}^{-1}(2p - 1) = t\,Q_{\mathcal{N}(0,1)}(p), \quad p \in (0,1),$$

*with the error function $\text{erf}$. Hence, (30) exactly becomes (not only in distribution)*

$$Z_t = (1 - t)Y_0 + t\,Q_{\mathcal{N}(0,1)}(U) = (1 - t)Y_0 + tZ,$$

*where $Z := Q_{\mathcal{N}(0,1)}(U) \sim \mathcal{N}(0,1)$, i.e. the components of (7) with the choice $\alpha_t = 1 - t$, $\sigma_t = t$. Furthermore, since $R_{t^2}(z) = R_{\mathcal{N}(0,t^2)}(z) = \frac{1}{2}(1 + \text{erf}\left(\frac{z}{t\sqrt{2}}\right))$, the quantile interpolant (12) reads as*

$$I_{s,t}(x,y) = (1 - s)x + s\sqrt{2}\,\text{erf}^{-1}\left(\text{erf}\left(\frac{y - (1 - t)x}{t\sqrt{2}}\right)\right) = (1 - s)x + \frac{s}{t}(y - (1 - t)x)$$

$$= ((1 - s) - \frac{s}{t}(1 - t))x + \frac{s}{t}y.$$

*which is exactly $\text{DDIM}_{s,t}(x,y)$ in (32) with $\alpha_t = f(t)$ and $\sigma_t^2 = g(t)$. $\diamond$*

Exactly as the DDIM interpolants, our quantile interpolants (31) satisfy the following crucial interpolation properties.

**Proposition 11** (a.k.a Proposition 4). *For all $x, y \in \mathbb{R}$ and all $s, r, t \in [0, 1]$, it holds*

$$I_{0,t}(x,y) = x, \quad I_{t,t}(x,y) = y, \tag{33}$$

*and*

$$I_{s,r}(x, I_{r,t}(x,y)) = I_{s,t}(x,y).$$

*Furthermore, inserting the quantile process (11) yields*

$$I_{s,t}(Z_0, Z_t) = Z_s. \tag{34}$$

*Proof.* By assumptions it holds

$$I_{0,t}(x,y) = f(0)x + Q_{g(0)}\big(R_{g(t)}(y - f(t)x)\big) = x,$$

and

$$I_{t,t}(x,y) = f(t)x + Q_{g(t)}\big(R_{g(t)}(y - f(t)x)\big) = y.$$

Furthermore, it holds the interpolation/consistency property

$$
\begin{aligned}
I_{s,r}(x, I_{r,t}(x,y)) &= f(s)x + Q_{g(s)}\big(R_{g(r)}(I_{r,t}(x,y) - f(r)x)\big) \\
&= f(s)x + Q_{g(s)}\big(R_{\cancel{g(r)}}(\cancel{f(r)x} + Q_{\cancel{g(r)}}\big(R_{g(t)}(y - f(t)x)\big) - \cancel{f(r)x})\big) \\
&= f(s)x + Q_{g(s)}\big(R_{g(t)}(y - f(t)x)\big) \\
&= I_{s,t}(x,y)
\end{aligned}
$$

for all $x, y \in \mathbb{R}$. Also note that inserting the random variables $Z_0, Z_t$ yields

$$
\begin{aligned}
I_{s,t}(Z_0, Z_t) &= f(s)Z_0 + Q_{g(s)}\big(R_{g(t)}(Z_t - f(t)Z_0)\big) \\
&= f(s)Z_0 + Q_{g(s)}(U) \\
&= Z_s.
\end{aligned}
$$

This finishes the proof. $\qquad\square$

Proposition 11 represents the key observation which allows us to utilize our quantile process (30) in the IMM framework the same way as Zhou et al. (2025) employ the DDIM interpolants (32):

For this, let us now recall the basic idea of inductive moment matching and the corresponding loss functions. Let us distinguish between real numbers written in small letters ($x_0, u, z_t \in \mathbb{R}$) and random variables written with capital letters ($X_0, U, Z_t, \ldots$). We assume that the probability distributions have densities:

| $\mathrm{Law}(X_0)$ | $\mathrm{Law}(Z_t)$ | $\mathrm{Law}(Z_s \mid X_0 = x_0, Z_t = z_t)$ | $\mathrm{Law}(Z_t \mid X_0 = x_0, U = u)$ | $\mathrm{Law}(X_0 \mid Z_t = z_t)$ |
|---|---|---|---|---|
| $\rho_0(x_0)$ | $\rho_t(z_t)$ | $\rho_{s\mid 0,t}(z_s \mid x_0, z_t)$ | $\rho_{t\mid 0,1}(z_t \mid x_0, u)$ | $\rho_{0\mid t}(x_0 \mid z_t)$ |

Note that by (34) we have $\rho_{s\mid 0,t}(z_s \mid x_0, z_t) = \mathrm{Law}(I_{s,t}(x_0, z_t))(z_s) = \delta(z_s - I_{s,t}(x_0, z_t))$, hence sampling from $\rho_{s\mid 0,t}(z_s \mid x_0, z_t)$ is just applying $I_{s,t}(x_0, z_t)$. Similarly, sampling from $\rho_{t\mid 0,1}(z_t \mid x_0, u)$ is just evaluating $I_{t,1}(x_0, Q_1(u))$.

The following proposition follows directly from Proposition 11 as in Zhou et al. (2025). It is essential for deriving the appropriate loss functions.

**Proposition 12.** *For all $0 \le s \le r \le t \le 1$, the quantile interpolant (31) is self-consistent, i.e.*

$$\rho_{s\mid 0,t}(z_s \mid x_0, z_t) = \int_{\mathbb{R}} \rho_{s\mid 0,r}(z_s \mid x_0, z_r)\, \rho_{r\mid 0,t}(z_r \mid x_0, z_t)\, \mathrm{d}z_r,$$

*and the quantile process (30) is marginal preserving, i.e.*

$$\rho_s(z_s) = \mathbb{E}_{z_t \sim \rho_t, x_0 \sim \rho_{0\mid t}(\cdot \mid z_t)}\big[\rho_{s\mid 0,t}(z_s \mid x_0, z_t)\big].$$

**Learning.** The conditional probability $\rho_{0\mid t}(\cdot \mid z_t)$ is now approximated by a network $p^\theta_{s,t,z_t}$ where the parameter $s$ describes the dependence on $\rho_s$ such that

$$\rho_s \approx \mathbb{E}_{z_t \sim \rho_t, x_0 \sim p^\theta_{s,t,z_t}}\big[\rho_{s\mid 0,t}(\cdot \mid x_0, z_t)\big] =: p^\theta(s,t). \tag{35}$$

Then it is proposed in (Zhou et al., 2025, Eq. (7)) to minimize the so-called *naïve objective*

$$\mathcal{L}_{\mathrm{naive}}(\theta) := \mathbb{E}_{s,t}\big[D(\rho_s, p^\theta(s,t)\big], \tag{36}$$

with an appropriate metric $D$, e.g. MMD. The procedure is now as follows: starting in a sample $x_0$ from $X_0$, we can sample $z_s, z_t$ from $Z_s, Z_t$ by (30), respectively; then given $z_t$ we sample $\tilde{x}_0$ from $p^\theta_{s,t,z_t}$, and finally we can evaluate $\tilde{z}_s = I(\tilde{x}_0, z_t)$ from (34), which is then compared with $z_s$.

**Inference.** The following iterative multi-step sampling can be applied: for chosen decreasing $t_k \in (0,1]$, $k = 0, \ldots, T$ with $t_0 = 1$, starting with $x_0^{(0)} \sim p_{0,1,z_1}^\theta$, we compute

$$z_{t_k} = I_{t_k, t_{k-1}}\left(x_0^{(k-1)}, z_{t_{k-1}}\right), \quad x_0^{(k)} \sim p_{0, t_k, z_{t_k}}^\theta, \quad k = 1, \ldots, T.$$

Although for marginal-preserving interpolants, a minimizer of $\mathcal{L}_{\text{naive}}$ exists with minimum 0, the authors of Zhou et al. (2025) object that directly optimizing (36) faces practical difficulties when $t$ is far away from $s$. Instead, they propose to apply the following "inductive bootstrapping" technique:

**Bootstrapping.** Instead of minimizing (36), we consider the *general objective*

$$\mathcal{L}_{\text{general}}(\theta) := \mathbb{E}_{s,t}\left[w(s,t)\text{MMD}_K^2(p^{\theta_{n-1}}(s,r), p^{\theta_n}(s,t))\right], \tag{37}$$

with a weighting function $w(s,t)$ to be chosen. The kernel $K$ of the squared MMD distance can be chosen as e.g. the (time-dependent) Laplace kernel. Importantly, the value $r$ is chosen to be a function $r = r_{s,t} \in [s,t]$ being "close to $t$" and fulfilling a suitable monotonicity property.

Let us assume the simplest case $r_{s,t} := \max\{s, t - \varepsilon\}$ with a small fixed $\varepsilon > 0$ and hereby demonstrate the bootstrapping technique: Fix $s \in [0,1]$. Then, it holds for all $t \in [s, s+\varepsilon]$ that $r_{s,s} = s$. By the definition (35) and property (33), it holds (independently of $\theta$) that $p^\theta(s,s)(z_s) = \rho_s(z_s)$. Hence, minimizing (37) in the first step $n = 1$ yields

$$0 = \text{MMD}_K^2(p^{\theta_0}(s,s), p^{\theta_1}(s,t_1)) = \text{MMD}_K^2(\rho_s, p^{\theta_1}(s,t_1)) \quad \text{for all } t_1 \in [s, s+\varepsilon].$$

In the second step $n = 2$, it holds for all $t_2 \in [s, s+2\varepsilon]$ that $r_{s,t_2} \in [s, s+\varepsilon]$. Hence, minimizing (37) in the second step yields, together with the first step,

$$0 = \text{MMD}_K^2(p^{\theta_1}(s, r_{s,t_2}), p^{\theta_2}(s,t_2)) = \text{MMD}_K^2(\rho_s, p^{\theta_2}(s,t_2)) \quad \text{for all } t_2 \in [s, s+2\varepsilon].$$

Thus, for the number of steps $n \to \infty$, it holds $0 = \text{MMD}_K^2(\rho_s, p^{\theta_n}(s,t_n))$ even for the entire interval $t_n \in [s, 1]$. Hence, minimizing the general objective (37) with a large number of steps eventually minimizes the naïve objective (36), see (Zhou et al., 2025, Theorem 1) for more details.

## D    ADAPTING NOISE TO DATA

### D.1    COUNTEREXAMPLE: MARGINAL PRODUCT

For the measure

$$\mu = \tfrac{1}{2}\delta_{(1,1)} + \tfrac{1}{2}\delta_{(-1,-1)} \in \mathcal{P}_2(\mathbb{R}^2), \qquad \mu_{\text{marg}} = \left(\tfrac{1}{2}\delta_{-1} + \tfrac{1}{2}\delta_1\right) \otimes \left(\tfrac{1}{2}\delta_{-1} + \tfrac{1}{2}\delta_1\right),$$

one has $W_2^2(\mu, \mu_{\text{marg}}) = 2$, whereas for

$$\nu_\alpha = \left(\tfrac{1}{2}\delta_{-\alpha} + \tfrac{1}{2}\delta_\alpha\right) \otimes \left(\tfrac{1}{2}\delta_{-\alpha} + \tfrac{1}{2}\delta_\alpha\right)$$

it holds $W_2^2(\mu, \nu_\alpha) = 2(1 - \alpha + \alpha^2) = 1.5$ for $\alpha = 0.5$. Thus the $W_2$–closest independent latent may contract or expand the marginals to partially account for correlations it cannot represent.

## D.2 DETAILS ON THE ARCHITECTURE OF THE LEARNED QUANTILES

We implement each one–dimensional quantile function with rational–quadratic splines (RQS) Gregory & Delbourgo (1982); Durkan et al. (2019). We explored several ways to map $u \in (0,1)$ into the spline input; the two variants below consistently performed well and are used in our experiments. For every coordinate $i$, we write

$$Q_\phi^i(u) \;=\; S_\phi^i(\psi(u)), \qquad u \in (0,1),$$

where $S_\phi^i : \mathbb{R} \to \mathbb{R}$ is a strictly increasing RQS with an interior knot interval $(-B, B)$ (with $K$ bins) and linear tails outside $\pm B$ that are $C^1$-matched at the boundaries. The two settings differ only in the "activation" $\psi$:

$$\text{(A) Logit: } \psi(u) = \text{logit}(u), \qquad \text{(B) Affine: } \psi(u) = \alpha_B(u) = B(2u - 1).$$

Thus, both (A) and (B) share exactly the same spline $S_\phi^i$ architecture—including the bounded interior $(-B, B)$ and slope-matched linear tails—and differ only in how $(0,1)$ is mapped into the spline's input. In (A), $\psi(u) \in \mathbb{R}$ and the linear tails of $S_\phi^i$ are used whenever $|\text{logit}(u)| > B$; in (B), $\psi(u) \in (-B, B)$ so the forward pass never touches the tails (they remain important for invertibility and out-of-range evaluation).

**Parameterization and constraints.**  Each spline $S_\phi^i$ is parameterized by raw bin widths, heights, and knot slopes. We pass these raw parameters through softplus, normalize widths and heights to sum to one (scaled to the domain span $2B$ and the learned range span, respectively), and add a small constant $s_{\min} > 0$ to each slope to enforce a positive lower bound. The linear tail slopes (left/right) are learned in the same way and are chosen so that both function value and slope agree at $\pm B$. These constraints guarantee strict monotonicity, hence $Q_\phi^i$ is strictly increasing on $(0,1)$ under both (A) and (B). Closed-form formulas for the spline pieces and their (log-)derivatives are available; by the chain rule,

$$\frac{d}{du}Q_\phi^i(u) = S_\phi^{i\prime}(\psi(u))\,\psi'(u), \quad \text{with} \quad \psi'(u) = \begin{cases} \frac{1}{u(1-u)} & \text{for (A)}, \\ 2B & \text{for (B)}. \end{cases}$$

**Per-component affine wrapper (scale/bias).**  After computing $Q_\phi^i(u)$, we add a tiny affine head per coordinate:

$$\tilde{Q}_\phi^i(u) \;=\; s_i\, Q_\phi^i(u) + b_i, \qquad s_i \;=\; \text{softplus}\big(\log \alpha_i\big), \quad b_i \;=\; \beta_i,$$

where $\alpha_i > 0$ and $\beta_i \in \mathbb{R}$ are learned per component. Using $\text{softplus}(\log \alpha_i)$ keeps $s_i > 0$ with a convenient dynamic range; this preserves monotonicity and adds only one scale and one bias parameter per component.

**Regularization via Expected Negative Log–Jacobian**  Let $Q_\phi : (0,1)^d \to \mathbb{R}^d$ be the componentwise map with affine heads, $Q_\phi(u) = \big(\tilde{Q}_\phi^1(u_1), \dots, \tilde{Q}_\phi^d(u_d)\big)$. Since the construction is per–coordinate, the Jacobian is diagonal with entries $\partial_{u_i}\tilde{Q}_\phi^i(u_i) > 0$. We regularize with the expected negative log–determinant of the Jacobian:

$$\mathcal{L}_{\text{reg}}(\phi) = \lambda_{\text{reg}}\, \mathbb{E}_{u \sim p_U}\big[-\text{logdet}\, J_{Q_\phi}(u)\big]$$

$$= \lambda_{\text{reg}}\, \mathbb{E}_{u \sim p_U}\Big[-\sum_{i=1}^{d} \log\big(\partial_{u_i}\tilde{Q}_\phi^i(u_i)\big)\Big].$$

Here $p_U = \text{Unif}\big((0,1)^d\big)$. In practice, we evaluate the log–derivatives in closed form.

**Computational efficiency and scalability.** The quantile architecture is highly efficient in both computation and memory. Each component $i$ requires only $\mathcal{O}(K)$ parameters for the RQS (where $K$ is the number of bins) plus two affine parameters, totaling roughly $4K + 2$ parameters per dimension for typical implementations. For a $d$-dimensional problem, this yields $\mathcal{O}(d \cdot K)$ total parameters—negligible compared to modern UNet architectures which often contain millions of parameters. Forward evaluation of $Q_\phi(u)$ involves $d$ independent spline evaluations operating in parallel. The diagonal Jacobian structure means that both the determinant and its gradient reduce to $d$ independent scalar derivatives with analytical closed-form expressions which are fully parallelizable, avoiding expensive automatic differentiation of matrix operations.

In practice, as noted in Section 5, the computational overhead (on CIFAR10) during joint training is approximately $3.2\%$ and drops to $1.2\%$ after freezing the quantile. Furthermore we only used 300k parameters for the quantile in contrast to 35M for the U-Net, making the approach highly scalable to high-dimensional problems. The strict monotonicity constraints and bounded parameterization (via softplus and normalization) ensure numerical stability throughout training, and we observed no instabilities across our experiments spanning dimensions from $d = 2$ to $d = 3072$ (CIFAR-10).

### D.3 TOY TARGET DISTRIBUTIONS

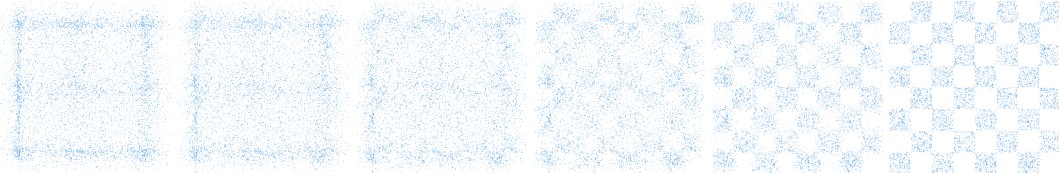

Figure 12: A generated sample path from the learned quantile latent to the checkerboard. The adapted latent (left) is already close to the target distribution.

We use three standard challenging low-dimensional distributions: Neal's funnel, a $3 \times 3$ Gaussian mixture, and a checkerboard.

**Funnel.** For the toy illustration in Figure 2, we work with the dataset known as Neals Funnel Neal (2003). The distribution of Neal's funnel is defined as follows:

$$p(x_1, x_2) \ = \ \mathcal{N}\big(x_1; 0, 3\big)\, \mathcal{N}\big(x_2; 0, \exp(x_1/2)\big).$$

**Grid Gaussian Mixture.** We give more details about the mixture of Gaussian we consider in our experiment. It is designed in a grid pattern in $[-1, 1]^2$, as follows:

$$\sum_{i=1}^{9} w_i \cdot \mathcal{N}(\mu_i, \sigma^2 I_2)\,,$$

where $(w_i)_{i=1}^9 = (0.01, 0.1, 0.3, 0.2, 0.02, 0.15, 0.02, 0.15, 0.05)$, $\mu_i = (\mu_1, \mu_2)$ with $\mu_1 = (i \bmod 3) - 1$, $\mu_2 = \left\lfloor \frac{i}{3} \right\rfloor - 1$, and $\sigma = 0.025$.

**Checkerboard.** Fix $\ell < h$ and domain $\Omega = [\ell, h]^2$. Define the support

$$\mathcal{S} = \big\{(x, y) \in \Omega : \ \lfloor x \rfloor + \lfloor y \rfloor \text{ is even}\big\}.$$

The checkerboard distribution is uniform on $\mathcal{S}$ and zero elsewhere:

$$
p_{\text{Checker}}(x, y) = \begin{cases} \dfrac{1}{\text{area}(\mathcal{S})}, & (x, y) \in \mathcal{S}, \\ 0, & \text{otherwise.} \end{cases}
$$

For integer $\ell, h$ with even side length (e.g. $\ell = -4, h = 4$), exactly half of $\Omega$ is active, hence

$$
p_{\text{Checker}}(x, y) = \frac{2}{(h - \ell)^2} \, \mathbf{1}_{\mathcal{S}}(x, y).
$$

### D.4 Loss Implementation

For training, the minibatch OT is computed empirically as follows: draw a minibatch $\{\mathbf{x}_0^{(i)}\}_{i=1}^B \sim \mu_0$ and $\{\mathbf{u}^{(j)}\}_{j=1}^B \sim \mathcal{U}([0, 1]^d)$, set $\mathbf{y}^{(j)} = \mathbf{Q}_\phi(\mathbf{u}^{(j)})$, and define the empirical measures

$$
\hat{\mu}_0^B = \frac{1}{B} \sum_{i=1}^B \delta_{\mathbf{x}_0^{(i)}}, \qquad \hat{\nu}_\phi^B = \frac{1}{B} \sum_{j=1}^B \delta_{\mathbf{y}^{(j)}}.
$$

The minibatch quantile alignment objective is

$$
\widehat{\mathcal{L}}_{\text{AN}}(\phi) \;=\; W_2^2\big(\hat{\mu}_0^B, \hat{\nu}_\phi^B\big),
$$

and gradients backpropagate through $\mathbf{y}^{(j)} = \mathbf{Q}_\phi(\mathbf{u}^{(j)})$. Let $T : \{1, \ldots, B\} \to \{1, \ldots, B\}$ denote the optimal assignment that minimizes $\sum_{i=1}^B \|\mathbf{x}_0^{(i)} - \mathbf{y}^{(T(i))}\|_2^2$, and define its inverse $P(j) = i$ such that $T(i) = j$. We use the conditional flow path $\mathbf{x}_t^{(j)} = (1 - t_j)\mathbf{x}_0^{(P(j))} + t_j \, \mathbf{y}^{(j)}$, $j = 1, \ldots, B$, with $t_j \sim \mathcal{U}(0, 1)$. The target velocity is $\mathbf{y}^{(j)} - \mathbf{x}_0^{(P(j))}$, we apply a stop-gradient operator $\text{sg}(\cdot)$ to this target in the flow matching loss. This prevents gradients from the velocity model from flowing back through the quantile function in this term, ensuring that $\mathbf{Q}_\phi$ is updated primarily through $\widehat{\mathcal{L}}_{\text{AN}}$, while $v_\theta$ learns to match the transport directions defined by the current quantile map. Note however the stop gradient operation only slightly stabilizes training, we can train with full gradients as well. We optimize the empirical version

$$
\widehat{\mathcal{L}}_{\text{CFM}}(\theta, \phi) = \frac{1}{B} \sum_{j=1}^B \big\| v_\theta\big(\mathbf{x}_t^{(j)}, t_j\big) - \text{sg}\big(\mathbf{y}^{(j)} - \mathbf{x}_0^{(P(j))}\big)\big\|_2^2, \quad \widehat{\mathcal{L}}(\theta, \phi) = \widehat{\mathcal{L}}_{\text{CFM}}(\theta, \phi) + \lambda \, \widehat{\mathcal{L}}_{\text{AN}}(\phi).
$$

## D.5 Algorithm

---

**Algorithm 1** Joint learning of 1D quantiles and FM velocity

---

**Require:** Dataset $\mathcal{D}$, batch size $B$, weight $\lambda$, iterations $K$
**Require:** Quantile model $\mathbf{Q}_\phi$, velocity model $v_\theta$
1: **for** $k = 1$ to $K$ **do**
2:      Sample $\{\mathbf{x}_i\}_{i=1}^B \sim \mathcal{D}$, $\{\mathbf{u}_j\}_{j=1}^B \sim \mathcal{U}([0,1]^d)$, $\{t_j\}_{j=1}^B \sim \mathcal{U}(0,1)$
3:      $C_{ij} \leftarrow \|\mathbf{x}_i - \mathbf{Q}_\phi(\mathbf{u}_j)\|_2^2$
4:      $T \leftarrow \arg\min_T \sum_{i=1}^B C_{i,T(i)}$
5:      Define $P$ by $P(j) = i$ such that $T(i) = j$
6:      $\hat{\mathbf{x}}_j \leftarrow \mathbf{x}_{P(j)}$
7:      $\mathbf{z}_j \leftarrow (1 - t_j)\hat{\mathbf{x}}_j + t_j \, \mathbf{Q}_\phi(\mathbf{u}_j)$
8:      $v_{\text{target},j} \leftarrow \text{sg}(\mathbf{Q}_\phi(\mathbf{u}_j) - \hat{\mathbf{x}}_j)$
9:      $\widehat{\mathcal{L}}_{\text{AN}} \leftarrow \frac{1}{B} \sum_{j=1}^B \|\hat{\mathbf{x}}_j - \mathbf{Q}_\phi(\mathbf{u}_j)\|_2^2$
10:      $\widehat{\mathcal{L}}_{\text{CFM}} \leftarrow \frac{1}{B} \sum_{j=1}^B \|v_\theta(\mathbf{z}_j, t_j) - v_{\text{target},j}\|_2^2$
11:      $\widehat{\mathcal{L}} \leftarrow \widehat{\mathcal{L}}_{\text{CFM}} + \lambda \widehat{\mathcal{L}}_{\text{AN}}$
12:      Update $(\theta, \phi)$ by a gradient step on $\widehat{\mathcal{L}}$
13: **end for**
14: **return** $(\theta, \phi)$

---

## E Implementation Details

We support baseline flow matching, optional quantile pretraining, and joint quantile+velocity optimisation. Pretraining fits the RQS transport before optionally freezing it; joint training updates both modules simultaneously. Once the quantile learning rate decays to zero we freeze its weights and continue optimising the velocity field only.

The coupling plans are calculated using the Python Optimal Transport package Flamary et al. (2021). For inference simulate the corresponding ODEs using the torchdiffeq Chen (2018) package. For all models we only used the batch size $128$ and learning rate $2e - 4$ for the velocities. We use Adam Kingma & Ba (2015) as the optimizer. The quantiles are parameterised by rational-quadratic splines as described in D.2, we set the minimum bin width and height to $1e - 3$ and the minimum slope to $1e - 5$. We could in principle stack mutiple RQS layers, however for all of our experiments we use one layer.

### E.1 Synthetic Examples

All models include a sinusoidal time embedding and SiLU activation functions. In these low dimensional settings we need no regularization and used $\lambda = 50$.

**Funnel.** For all models we used 3 hidden layers with width 64. We used a batch size of 128, a learning rate of $2e - 4$ and exponential moving average on the network weights of $0.999$. The baselines were trained for 200,000 iterations. Since there is a very high variance when sampling from the funnel, we pretrain our quantiles and use the frozen quantiles during flow matching. We trained our quantile for 50,000 steps and to compensate we trained our velocity for only 100,000 steps. For the RQS we chose logit activation, 32 bins and a bound of 500.

**Grid Gaussian Mixture and Checker.** The quantiles were trained for the first 20,000 steps, after which the learning rate was linearly decayed to 0 by step 25,000. For both datasets, we trained the velocity model with 3 layers and a hidden width of 256 for 100,000 steps. Furthermore we used sinusoidal positional embeddings for the checkerboard. We found both bounded and logit activation performed well, for the RQS we chose 32 bins with a bound of 50.

### E.2 Image Experiments

For both image datasets, we adapt the U-Net from Nichol & Dhariwal (2021) to parametrize our velocity field.

**MNIST.** For the MNIST dataset we use the U-Net with channel multipliers $(1, 2, 4)$, two residual blocks per resolution, attention at $7 \times 7$, and 1 attention head. We clip the gradient norm to 1 and use exponential moving averaging with a decay of 0.99. We test three configurations with base widths of 8, 16, and 32 channels. For these ablation runs, we use quantile loss weight $\lambda = 1.0$, regularization parameter $\beta = 0.1$, and rational quadratic spline with 16 bins and bound 3.0. The quantiles were trained for the first 20,000 steps, after which the learning rate was linearly decayed to 0 over the next 10,000 steps. The images in Figure 5 were generated using our 32 channel configuration.

**CIFAR.** We use exactly the same U-Net setup from Tong et al. (2024). We clip the gradient norm to 1 and use exponential moving averaging with a decay of 0.9999. To evaluate our results, we use the Fréchet inception distance (FID) Heusel et al. (2017). The quantiles were trained for the first 20,000 steps, after which the learning rate was linearly decayed to 0 by step 25,000. We used $\lambda = 1$ and varied $\beta$. For the RQS we used logit activation, 32 bins and a bound of 25.

CIFAR-10 inputs are normalized to $[-1, 1]$ with random horizontal flips.

## F Further Experimental Results

### F.1 Synthetic Datasets

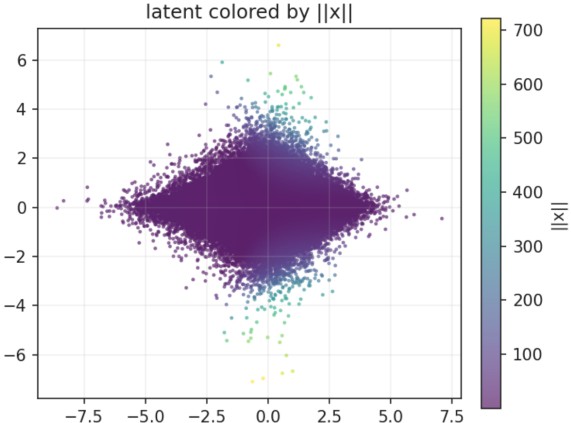

Figure 13: Samples (1M) from our learned latent of the funnel distribution. Color shows endpoint norm.

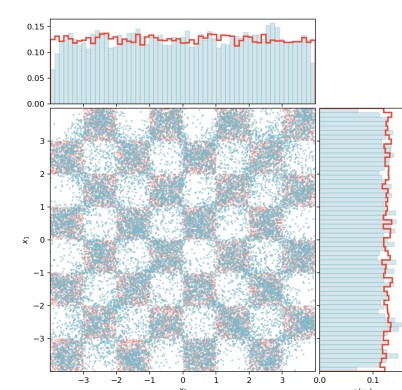 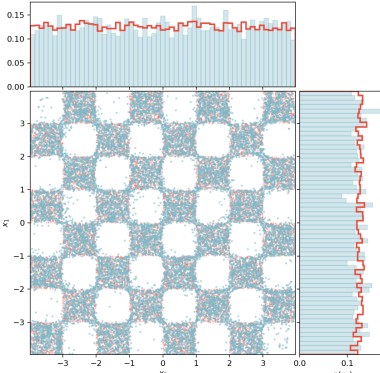

Figure 14: Flow Matching with optimal coupling using Gaussian noise (left) and our learned noise (right) after **20k** training steps with identical parameters. Generated samples are shown in blue, and ground-truth samples in red

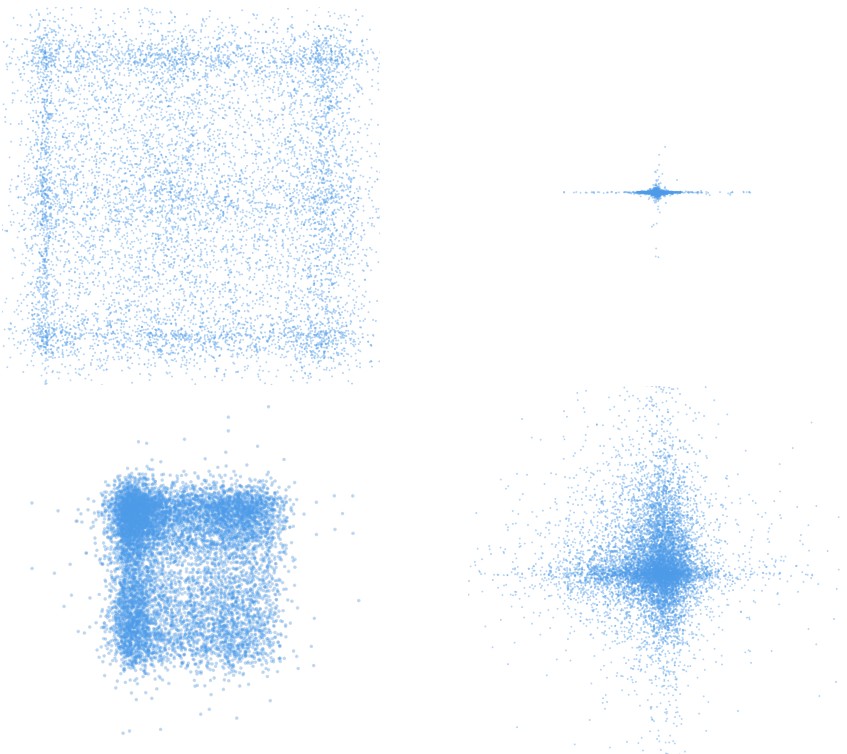

Figure 15: Visualization of the effect of the loss weight $\lambda$, on the left the learned latents for the checkerboard and the gaussian mixture example using $\lambda = 50$. On the right the learned latents using $\lambda = 0$. Without the additional loss the model tries to make predicting the velocity as simple as possible, this does not align with our objective.

## F.2   MNIST

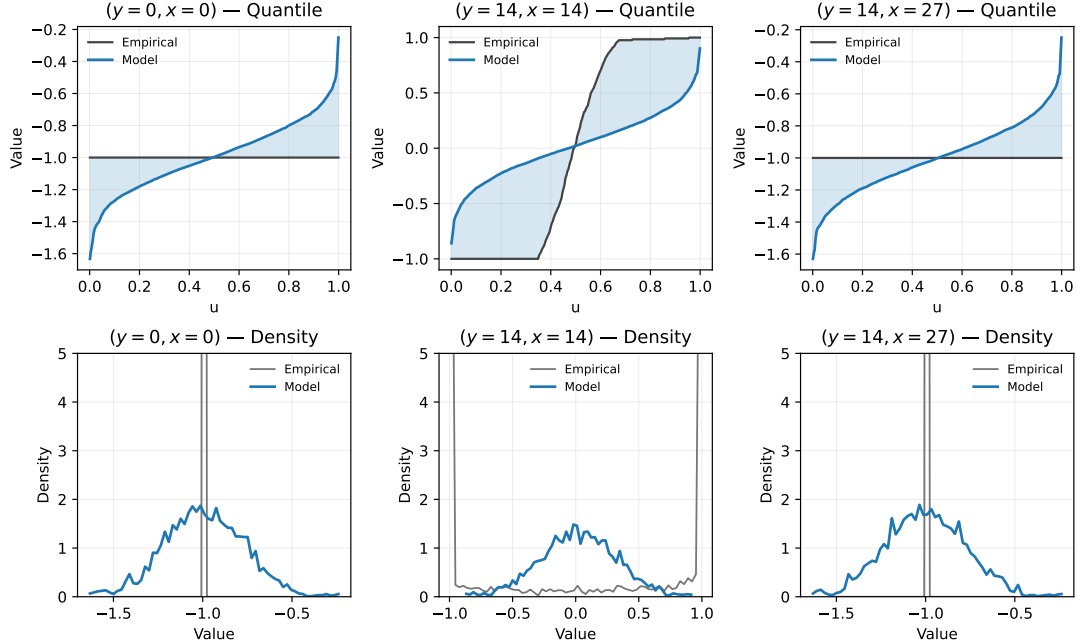

Figure 16: Comparison of the empirical and learned probability density functions and their quantile functions at different pixel locations $(y, x)$, averaged over images from the MNIST dataset. The blue area illustrates the difference between the quantiles, corresponding to the one-dimensional Wasserstein distance; see Eq. 4.1.

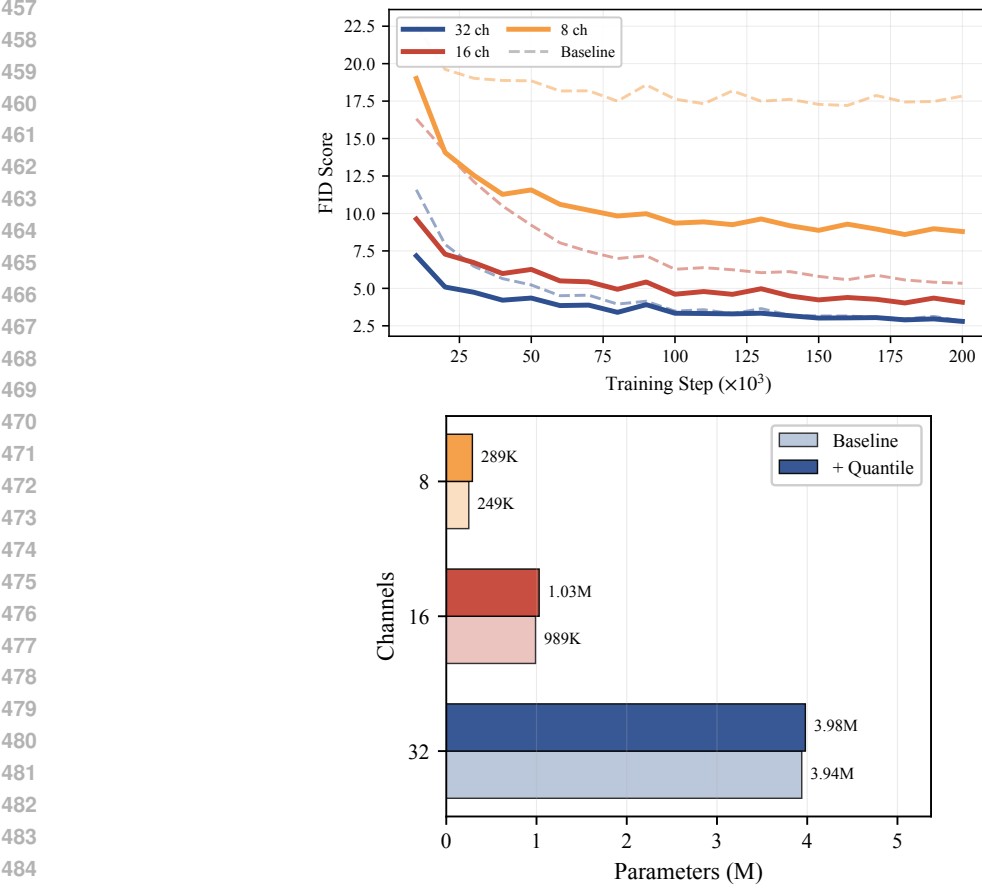

Figure 17: Ablation study over capacity of the U-Net for sampling from the MNIST dataset. The FID curves show that our method achieves significantly lower FIDs for lower capacities. Note the difference in parameters is approximately 40k.

## F.3 CIFAR10

| $\beta$ | FID (20 steps) | FID (100 steps) |
|---|---|---|
| 0.1 | $8.93_{\pm 0.04}$ | $5.22_{\pm 0.02}$ |
| 0.2 | $7.81_{\pm 0.04}$ | $4.75_{\pm 0.02}$ |
| 0.3 | $\mathbf{7.48}_{\pm 0.05}$ | $4.53_{\pm 0.05}$ |
| 0.4 | $7.60_{\pm 0.05}$ | $4.54_{\pm 0.01}$ |
| 0.5 | $7.66_{\pm 0.03}$ | $4.49_{\pm 0.02}$ |
| 0.6 | $7.70_{\pm 0.05}$ | $4.47_{\pm 0.03}$ |
| 0.7 | $7.93_{\pm 0.05}$ | $4.59_{\pm 0.02}$ |
| 0.8 | $7.77_{\pm 0.05}$ | $\mathbf{4.42}_{\pm 0.02}$ |
| 0.9 | $8.09_{\pm 0.04}$ | $4.56_{\pm 0.02}$ |
| 1.0 | $8.35_{\pm 0.03}$ | $4.66_{\pm 0.04}$ |
| 1.1 | $8.60_{\pm 0.04}$ | $4.80_{\pm 0.03}$ |
| Baseline | $8.42_{\pm 0.07}$ | $4.63_{\pm 0.05}$ |

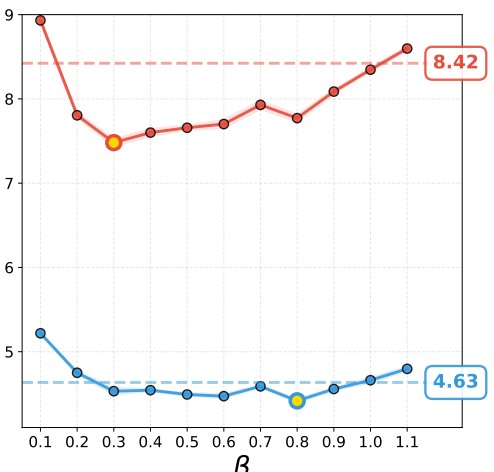

Figure 18: Complete FID scores on CIFAR-10 for all $\beta$ values. Our method reached the best validation FID after 320k steps, while the baseline took 340k. We used those checkpoints for the evaluation. We evaluated the FID using 5 seeds and report the mean as well as the standard deviation. Red denotes 20 step FID, blue 100 step FID, dotted line refers to baseline.

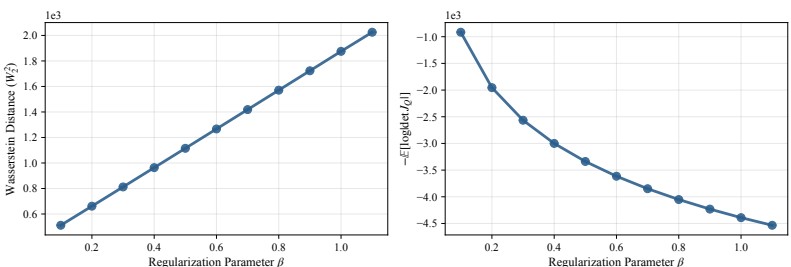

(f) Final performance metrics at 55k training steps as a function of regularization parameter $\beta$.

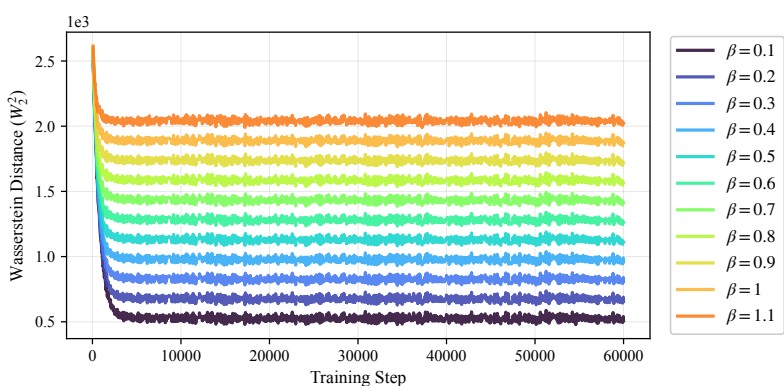

(b) Wasserstein distance evolution during training for different regularization parameters $\beta$.

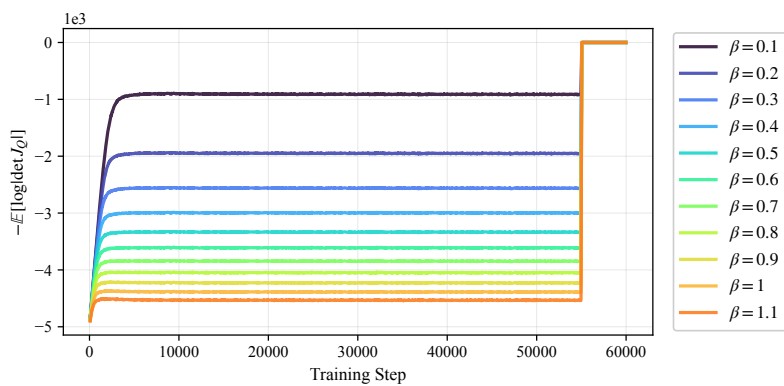

(c) Regularization loss $-\mathbb{E}[\log|\det J_Q|]$ showing the regularization loss across different $\beta$ values.

Figure 19: Ablation studies for regularization parameter $\beta$ and model capacity.

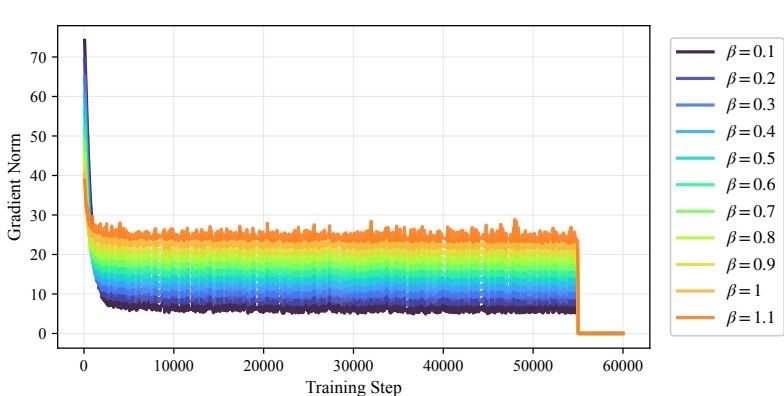

(d) Gradient norm of the quantile function during training.

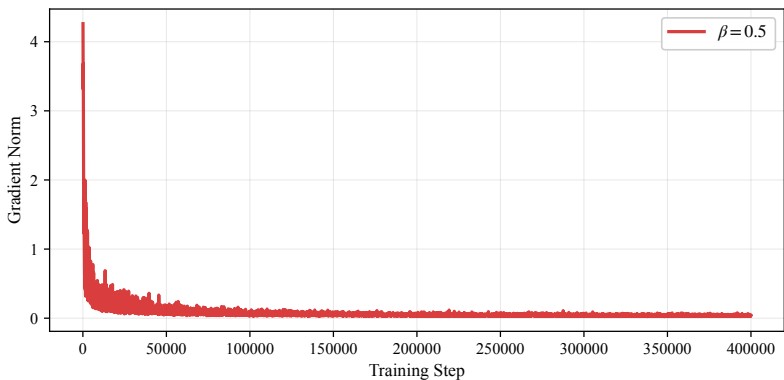

(e) Gradient norm of the velocity field for fixed $\beta = 0.5$ over training.

Figure 20: Ablation studies showing the effect of regularization and model capacity on training dynamics and final performance.

