# OpenReview forum: "Adapting Noise to Data: Generative Flows from learned 1D Processes"
_ICLR.cc/2026/Conference — Submitted to ICLR 2026_

### Official Review · Reviewer_Ghhr · 2025-10-15

**Soundness:** 3
**Presentation:** 2
**Contribution:** 3
**Rating:** 4
**Confidence:** 4

**Summary:**

The authors propose a method to learn the initial distribution of a flow matching model. Usually, this distribution is chosen to be Gaussian noise. Here, they parameterize the initial distribution with a learnable quantile function. Beyond the initial distribution, this distribution also determines all intermediate distributions (commonly called probability path), while having a fixed linear interpolant. They jointly learn this initial distribution with the velocity field. Experiments on synthetic data sets and small-scale image datasets show the validity of the method.

**Strengths:**

- This is a natural idea that has not been explored in the literature as much and that could be a powerful way of improving flow-based generative models.
- It is a simple training objective that has minimal computational overhead.

**Weaknesses:**

- Overall, the writing of the paper could be improved significantly. The motivation is not well-explained in the text (both in the introduction and later in the text). Further, illustrations and examples are lacking.
- The experiments are limited and the presented results are not very strong. For example, for CIFAR10, the flow baseline is worse than standard baselines (https://github.com/facebookresearch/flow_matching). FM achieves an FID on CIFAR10 <=3.0.
- The training objective requires more elaboration: The parameters phi that parameterize the initial distribution underlie a trade-off: They can be either used to  minimize the first or second term in the training objective L(theta, phi). Therefore, even for lambda=0, minimizing this objective might be valid (i.e. one minimizes then effectively the residual variance of the CFM loss). As such discussions are at the core of the idea, it would be good to elaborate on this more.

**Questions:**

- L26: "Consistency models like the recently introduced inductive moment matching (IMM) Zhou et al. (2025)" → Consistency Models are generally speaking different from IMM. I would rather present them as different methods.
- Proposition 2 is known prior to this work, e.g. it is a special case of Proposition 4 in [1] and should be referenced.
- Why are Consistency Models discussed? They are not really used anywhere?

---

> ### Author Response · Authors · 2025-11-19
>
> Many thanks for your feedback and the valuable suggestions.
> We have substantially revised our paper accordingly, placing a strong emphasis on clearer writing, and improved experiments. We give detailed answers to all your concerns below.
>
> **Weaknesses**
>
> >Overall, the writing of the paper could be improved significantly.[...]
>
> We have substantially revised the manuscript to address these concerns:
>
> - **Restructured preliminaries:** Section 2 now has clear subsections improving logical flow and accessibility.
>
> - **Added optimal transport coupling discussion:** Added Remark 2 on OT-FM, to contextualize our approach.
>
> - **Unified quantile framework:** Reorganized Sections 3-4 to provide clearer narrative, moving quantile processes/ interpolants into a unified Section 4 "Adapting Noise to Data" containing both the theoretical background (4.1) and learning methodology (4.2).
>
> - **Restructured experiments:** Rewrote Section 5 for significantly improved clarity and readability.
>
> >The experiments are limited and the presented results are not very strong.[...]
>
> We have updated our network architecture and CIFAR-10 experiments. The results are now in line with recent flow matching papers such as [1]. We acknowledge that further optimization (and especially larger networks) could improve results; however, our focus is on demonstrating the principle of learning adapted noise rather than achieving state-of-the-art image generation.
>
> Furthermore, we added the Figure 17 showing that our learned latent achieves significantly lower FID scores when using smaller network capacities on MNIST, demonstrating that removing redundant information enables more efficient parameter usage.
>
> >The training objective requires more elaboration:[...]
>
> We agree that this section could be improved. For clarity we updated Section 4.2 and Section D.4 in the appendix to outline the training procedure in more detail. We also updated the algorithm in the appendix (D.5). Furthermore we included Figures 19,20 in the appendix to outline the training dynamics.
>
> We would like to note the Wasserstein term $\mathcal{E}(\phi) = W_2^2(\mu_0, \nu_\phi)$ provides a direct gradient signal for $\phi$ that is independent of the velocity network $v_\theta$. This ensures that the latent captures marginal structure. We have added a visualization of the learned latent when we set $\lambda =0$ (Figure 15), also note this severely affects sampling performance.
>
> **Questions**
> > [...]Consistency Models are generally speaking different from IMM.[...]
>
> We thank the reviewer for this clarification. We understand that different communities may use "consistency models" to refer specifically to the Song et al. (2023) framework. In our manuscript, we use "consistency models" in a broader sense to refer to any generative model that learns a direct one-step mapping from noise to data. We have clarified this in the introduction.
>
> >Proposition 2 is known prior to this work, e.g. it is a special case of Proposition 4 in [1] and should be referenced.
>
> Thanks! Unfortunately your review does not contain any reference [1]. So we guess that you probably meant
>
> Generator Matching: Generative modeling with arbitrary Markov processes, P. Holderrieth et. al
>
> We have now added this reference, thank you! We want to clarify that this proposition is not a contribution of this paper but rather a known result that we leverage as a building block.
>
> >Why are Consistency Models discussed? They are not really used anywhere?
>
> The discussion of Consistency Models (Section 3.3 and the appendix on IMM) demonstrates that our quantile interpolants framework is general enough to extend beyond flow matching to other generative modeling paradigms. Specifically, we show that our quantile interpolants satisfy the same consistency properties as DDIM interpolants.
>
> While we do not provide experimental results with consistency models (due to space and focus), we include this theoretical connection to: (1) demonstrate the generality of our quantile process framework, and (2) provide a foundation for future work applying learned noise to consistency models.
>
> We agree that it had a too prominent role and we have reduced the discussion in the main body, moving details to the appendix.

---

> > ### Comment · Reviewer_Ghhr · 2025-11-22
> >
> > Thank you for the revised manuscript and providing answers to my questions. I read revised version and the new version is easier to understand, indeed. However, my main concern - a lack of clear experimental evidence for the suitability of the proposed method including benchmarks - is still lacking. CIFAR10 is saturated benchmark and the results are not better than the ones for standard FM (only if the capacity of the neural network is constrained to less channels, see figure 05). It would greatly strengthen to work to focus on a more challenging benchmark or showcase at least one empirical advantage of the proposed method compared to previous methods.

---

> ### Author Response · Authors · 2025-11-25
>
> Many thanks for acknowledging the improved clarity of the revised version. We were surprised about the comments on the numerical part and would like to **clarify possible misconceptions**.
>
> **Your concern states:**
> >"the results are not better than the ones for standard FM (only if the capacity of the neural network is constrained to less channels, see figure 05)."
>
> **This is not correct.** The capacity-constrained experiments in Figure 4 (previously Figure 5) are for **MNIST**, not CIFAR-10. Note we changed the order of Figure 4 and 5 to improve clarity.
>  On **CIFAR-10**, we achieve **0.2 FID improvement over standard FM at full network capacity**. This improvement is consistent with gains reported in recent flow matching papers [1,2], using the same experimental setup as [2].
>
> You may have overseen that our experiments demonstrate **multiple concrete advantages**:
>
> - **Heavy-tailed distributions** (Funnel, Fig. 2, 13): Our learned noise achieves **significant** improvements over standard FM with a Gaussian latent, and outperforms even the use of a well chosen heavy-tailed Student-t latent [3]. Heavy-tailed generation is a *challenging task* even in lower dimensions, see also [3,4,5].
>
> - **Clear marginal structure** (MNIST, Fig. 4, 5, 16, 17): At constrained capacity, our learned latent achieves substantially better FID by removing redundant marginal information—particularly relevant for capacity-limited applications where the data has clear marginal structure.
>
> - **Shorter transport paths** (Checkerboard, Fig. 1, 14): Learned noise can drastically reduce transport distance and accelerate training convergence.
>
> - **Scalability with minimal overhead**: Our method scales to high-dimensional problems. Even on CIFAR-10—a dataset *without* obvious marginal structure—we **improve** FID by 0.2 points with negligible computational cost  (Section D.2).
>
> **Our contribution:** We introduce a completely new methodology for designing latent distributions in generative models. We provide the motivation, theoretical foundation, practical implementation, and demonstrate how it works across diverse scenarios. We believe this provides a meaningful foundation that future work can build upon.
>
> Given our **extensive updates**, in which we also addressed your previous concerns, we would kindly ask you to reconsider your score.
>
> [1] AlignFlow: Improving Flow-based Generative Models with Semi-Discrete Optimal Transport, Kong et al.
>
> [2] Improving and generalizing flow-based generative models with minibatch optimal transport, Tong et al.
>
> [3] Heavy-Tailed Diffusion Models, Pandey et al.
>
> [4] Concentration of Measure for Distributions Generated via Diffusion Models, Ghane et al.
>
> [5] On the Statistical Capacity of Deep Generative Models, Tam and Dunson

---

### Official Review · Reviewer_vyUw · 2025-10-27

**Soundness:** 2
**Presentation:** 2
**Contribution:** 2
**Rating:** 2
**Confidence:** 3

**Summary:**

This paper introduces a novel generative modeling framework that constructs flow-based models using learned one-dimensional noising processes. Instead of relying on a fixed Gaussian latent distribution, the method learns the noise distribution directly through quantile functions that adapt to the data. This formulation integrates naturally with the flow matching framework, enabling more flexible and data-dependent noise modeling. The authors further illustrate the approach through several examples of one-dimensional processes, including the Wiener process, the Kac process, and an MMD gradient flow, and show that learning quantile-based noise can substantially enhance the flexibility and transport efficiency of generative models.

**Strengths:**

1. The idea of constructing generative flows through learnable 1D quantile processes is original.

2. By using quantile parameterizations, the approach can handle distributions with compact support or heavy tails, going beyond the Gaussian assumptions typical in flow and diffusion models.

3. The framework is compatible with standard objectives such as Flow Matching and Inductive Moment Matching, showing practical extensibility.

**Weaknesses:**

1. Lack of sufficient baselines.

The paper lacks adequate baseline comparisons to clearly demonstrate the advantages of the proposed method. In Section 5.1, no baseline is provided for reference, and Figure 5 includes only a single baseline whose selection and description are not well explained. The experimental evaluation should include more detailed quantitative comparisons against standard diffusion or flow-based models to better substantiate the claimed improvements.

2. Clarity and presentation issues.

The overall clarity of the paper can be improved. The abstract does not effectively summarize the key contributions and contains some redundancy. For example, the first and third sentences are quite similar. Several methodological details are also unclear. For instance, the statement “we pre-train our quantile” (Line 356) does not explain why pre-training is necessary or which experiments rely on it. Similarly, the introduction of the regularization term that penalizes the expected negative log-determinant of the Jacobian (Line 374) is mentioned without justification or analysis of its impact. These elements should be clarified to improve the transparency and reproducibility of the work.

3. Expressive power of one-dimensional processes.

The paper does not provide sufficient theoretical or empirical evidence regarding the expressive power of using one-dimensional denoising processes. While the decomposition into independent one-dimensional components makes the approach more tractable, it may limit the model’s ability to capture complex dependencies across dimensions. A deeper discussion or ablation study evaluating this trade-off would strengthen the paper’s technical soundness.

**Questions:**

1. Generality of one-dimensional flows

Is there a universal or systematic way to construct one-dimensional flows, beyond the three specific examples discussed in Section 4.1?

2. Sampling efficiency

What is the sampling time or computational cost of the proposed method compared to standard flow matching or diffusion-based models?

---

> ### Author Response · Authors · 2025-11-19
>
> Many thanks for the constructive feedback and the helpful suggestions. We have improved the paper accordingly and clarified several points to improve both presentation and technical transparency.
> Below we address all your comments in detail.
>
> **Weaknesses**
> >Lack of sufficient baselines.
>
> We have significantly improved our experimental section. We have updated our network architecture to match the setup used in [1]. We note that using a Gaussian latent is the standard setup for flow matching, making it a natural baseline for our approach. Since our contribution is learning an adapted noise distribution, comparing against this standard Gaussian baseline directly demonstrates the benefit of our method.
>
> >Clarity and presentation issues.
>
> Based on your review, we have significantly improved the clarity of the paper. The key improvements include:
> - Updated the abstract/introduction and reordered Sections 3 and 4 to better highlight the significance of our contributions.
> - Improvement of the experimental section:
>     - Generally improved clarity and writing.
>     - We added a more detailed discussion of the design choices, including mini-batch optimal coupling and log-determinant regularization.
>     - Added MNIST experiments showing the effect of network capacity constraints.
>     - Improved the CIFAR-10 experiment and added further ablations in the appendix.
>     - In the implementation details we clarify why we pretrain the quantile network on the funnel distribution: due to the extremely        high variance when sampling a minibatch from the funnel.
>
>    - Added a detailed explanation of the parametrisation of the quantiles and the regularization term in the appendix.
>
> >Expressive power of one-dimensional processes.
>
> You raise an important point about expressiveness. We want to clarify: our componentwise quantiles, by construction, cannot capture cross-dimensional dependencies. However, this is also true for standard FM and diffusion models, which rely on isotropic Gaussian noise $\mathcal{N}(0, I_d)$.
>
> The key insight is that in generative modeling, the *velocity field* (or score function in diffusion) is responsible for learning correlations and dependencies. Our learned noise $\nu_\phi$ is trained to be the closest factorized distribution to the data $\mu_0$ in Wasserstein distance. This means:
>
> - The quantile captures *marginal structure* (scales, tails, support)
> - The velocity field focuses on *cross-dimensional correlations*
> - By starting closer to the data, the velocity field has an easier learning problem and yields shorter paths
>
> This division of labor is demonstrated in our capacity ablation experiments on MNIST: smaller velocity networks maintain good performance with learned noise but degrade with Gaussian noise, suggesting the learned latent reduces the expressiveness requirements on the velocity field. See also the new Figure 4.
>
> **Questions**
>
> >Is there a universal or systematic way to construct one-dimensional flows, beyond the three specific examples discussed in Section 4.1?
>
> Thank you for this excellent question—it highlights a key strength of our framework. The reduction to 1D flows provides remarkable flexibility in design. There are multiple systematic approaches:
>
> **Approach 1 (Density-based):** Specify the 1D flow via probability densities $p_t$ that converge to $\delta_0$ as $t \to 0$. For example, one could:
>
> - Define a valid density $p_t$; there are many approaches for this
> - Compute the velocity field from the continuity equation $\partial_t p_t + \partial_x(p_t v_t) = 0$ by solving for $v_t$ (trivial in 1D since divergence is just a spatial derivative)
> - Sampling from 1D densities is generally straightforward
>
> **Approach 2 (Process-based):** Start with any 1D stochastic process $(Y_t)_t$ with $Y_0 = 0$. The velocity field is obtained as the conditional expectation: $v_t(x|0) = \mathbb{E}[\dot{Y}_t | Y_t = x]$ (this is the standard approach in stochastic analysis).
>
> >What is the sampling time or computational cost of the proposed method compared to standard flow matching or diffusion-based models?
>
> We can separate the complexity into two parts:
>
> **Training:** Our method is approximately 3.2% slower during the 55k iteration joint training phase and 1.2 % after freezing the quantile, resulting in only 1.5% total overhead over the full training (measured on CIFAR-10). The quantile network is very lightweight ($\mathcal{O}(d \cdot K)$ parameters) compared to the UNet.
>
> **Inference:** Once trained, sampling from our quantiles is extremely efficient:
>
> - RQS evaluation is essentially just a bin lookup followed by a quadratic interpolation
> - Because of the componentwise construction, all $d$ components are evaluated in parallel
> - The main sampling cost remains the ODE solver for the velocity field (same as standard FM)
>
> [1] Improving and generalizing flow-based generative models with minibatch optimal transport, Tong et. al

---

> ### Author Response · Authors · 2025-11-28
>
> Dear Reviewer vyUw,
>
> Thank you again for your review. We have carefully revised our paper and answered your questions to address the concerns you raised. We believe these changes significantly strengthen our work.
>
> We would greatly appreciate if you could review our responses and reconsider your score.
>
> Of course, if you have any further questions or concerns, we are happy to address them.

---

### Official Review · Reviewer_ySRN · 2025-10-31

**Soundness:** 3
**Presentation:** 2
**Contribution:** 2
**Rating:** 6
**Confidence:** 2

**Summary:**

The authors propose a framework for 1D per-dimension noising processes for generative
models. They propose to learn the latent distribution to reduce the transport paths of
generative models. The latent distribution is modeled via learned quantile functions, which
are modeled via rational quadratic splines. The quantile functions are learned from data by
minimizing the Wasserstein-2 distance between the data distribution and the modeled latent
distribution.

Main contribution:
- Decomposition of multidimensional flows into 1D noising processes
- Quantile-based formulation of latent distribution: learn the quantile function of latent
noise instead of fixing it to a Gaussian distribution
- Experimental validation on synthetic data (checkerboard, funnel, Gaussian mixture)
and image data (MNIST, CIFAR-10)

**Strengths:**

- Minimal computational overhead via rational quadratic splines
- improved noise distribution adapted to target
- Good explanation of the math fundamentals
- provides a general framework for independent 1D noising processes and an
expressive way to parameterize them in practice via quantile functions and rational quadratic splines

**Weaknesses:**

- Missing ablation study on the velocity not exploding outside of the support of the
distribution. A simple 2D example showing the vector field would be nice.
- Missing benchmarks on larger problems -> how scalable and stable is this approach
with an increasing problem dimension? -> potentially unstable quantile training
- lack of quantitative metrics
- unclear generalization capability for shifting data distributions

**Questions:**

The weight on the quantile loss and the regularization weight are both new
hyperparameters that need to be tuned. How sensitive are they to different problems? On the funnel target, the quantiles are pre-trained -> another hyperparameter.
- How stable is the joint optimization of quantile and flow networks in practice?
- Is learning quantiles equivalent to learning transport maps under certain
assumptions?
- How does the learned quantile noise compare to learned latent priors in VAEs or
normalizing flows?
Can this method scale to modern high-resolution diffusion tasks?
- What happens for out-of-distribution conditional data?

**Details Of Ethics Concerns:**

None.

---

> ### Author Response · Authors · 2025-11-19
>
> Many thanks for the elaborate feedback and suggestions for additional experiments. We have addressed your concerns in the revision.
> Here are the detailed answers.
>
> **Weaknesses**
> >Missing ablation study on the velocity not exploding outside of the support of the distribution[...]
>
> This is a very good point, the diffusion example demonstrates that velocity fields may explode outside the support of e.g. target Dirac points. Luckily, our quantile framework allows for an exact description of the velocity's $L_2$ norm via the $L_2$ norm of the quantile function. Hence, explosions of the velocity field can be avoided by suitable contraints in the quantile's parameterization. We added a suitable remark in the appendix A4.
>
> >Missing benchmarks on larger problems -> how scalable and stable is this approach with an increasing problem dimension?[...]
>
> We have added extensive discussion of scalability and stability:
>
> **Scalability:** As detailed in the appendix on learned quantiles, our approach scales well to high dimensions due to:
> - Linear parameter scaling: $\mathcal{O}(d \cdot K)$ parameters for $d$ dimensions and $K$ spline bins
> - Parallel computation: All $d$ quantile components are evaluated independently
> - Minimal overhead: only 1.5% total overhead over 400k iterations (measured on CIFAR-10 with $d=3072$)
>
>
> **Stability:** In our experiments across synthetic and image datasets, we observe stable joint training without special initialization or learning rate tuning. The quantile network typically converges within a fraction of the total iterations after which we freeze it and continue training only the velocity network. Also see Figures 19,20 in the appendix.
>
> >lack of quantitative metrics
>
> We have added the following quantitative evaluations:
> - New FID scores on CIFAR-10 over a wider range of regularization parameters
> - Capacity ablation on MNIST showing FID vs. network size
> - Ablation on the effect of the regularization parameter on CIFAR10 (appendix)
>
> >unclear generalization capability for shifting data distributions
>
> Thank you for raising this important point. We want to clarify what our method is able to do concerning distribution shifts:
>
>  Our quantile functions are trained to minimize the Wasserstein distance to the training data distribution. If the data distribution shifts, the quantile would need to be retrained or updated. However as we further emphasize, this retraining is computationally very inexpensive.
>
> **Questions**
> >The weight on the quantile loss and the regularization weight are both new hyperparameters that need to be tuned. How sensitive are they to different problems? On the funnel target, the quantiles are pre-trained -> another hyperparameter.
>
> We found the hyperparameters to be robust across the datasets we tested. In our experiments, varying them within reasonable ranges consistently yielded good performance without requiring extensive problem-specific tuning.
> For the funnel distribution, pretraining was used as a practical choice due to the high variance when sampling minibatches from this target, rather than due to instability in the training procedure itself. In general, we found the hyperparameters straightforward to select and observed that the method was not highly sensitive to these choices.
>
> >How stable is the joint optimization of quantile and flow networks in practice?
>
> We refer to our answer on stability above. Especially with respect to the joint optimization the new Figures 19,20 in the appendix highlight the stability during joint training.
>
> >Is learning quantiles equivalent to learning transport maps under certain assumptions?
> In the 1D case, yes: for any probability measure $\mu$ on $\mathbb{R}$, its quantile function $Q_\mu$ is exactly the optimal transport map from $\mathcal{U}(0,1)$ to $\mu$ (a classical result in optimal transport theory).
>
> However, in our $d$-dimensional setting, we learn *componentwise* (factorized) quantiles $Q_\phi = (Q^1_\phi, \ldots, Q^d_\phi)$. This means we learn a product measure $\nu_\phi$ that is close to the data distribution $\mu_0$ in Wasserstein distance, but generally $\nu_\phi \neq \mu_0$ due to independence (see appendix D.1). Our quantiles learn an adapted noise distribution capturing marginal structure and reducing transport cost, while the velocity network $v_\theta$ handles cross-dimensional correlations.
>
> >How does the learned quantile noise compare to learned latent priors in VAEs or normalizing flows?[...]
>
> While learned VAE/flow priors usually model joint latent distributions, our factorized quantile functions learn only componentwise marginals with negligible overhead delegating dependencies to the velocity field.
>
> >What happens for out-of-distribution conditional data?
>
> Our current work focuses on unconditional generation. Extending to conditional generation (class-conditional, text-to-image, etc.) is an important direction for future work.

---

> > ### Comment · Reviewer_ySRN · 2025-11-27
> > **Reply to authors**
> >
> > I thank the authors for their reply. I will maintain my score.

---

> ### Author Response · Authors · 2025-11-28
>
> Dear Reviewer ySRN,
> we guess your short comment means that we have addressed all your concerns.
> Is there any reason why you do not increase the score?

---

### Official Review · Reviewer_v4Fw · 2025-11-05

**Soundness:** 3
**Presentation:** 4
**Contribution:** 4
**Rating:** 10
**Confidence:** 4

**Summary:**

This paper introduces an approach that uses one dimensional processes and quantile functions to learn generative models in a component-wise manner. The author shows how this approach is compatible with the flow matching and consistency model frameworks, and can better handle difficult settings such as heavy tails and compact supports.

**Strengths:**

Originality: The core ideas (e.g., using 1D processes and quantile functions to learn generative models in a way that is compatible with consistency and flow matching frameworks, etc) are, to the best of my knowledge, original and innovative.

Clarity: the paper is written in a clear and self-contained manner. Even in the more technical portions, everything is defined and explained clearly. This is a major strength of the paper.

Quality: I find the quality of the theoretical and empirical sections to be sufficient. While one can always perform more experiments on more datasets/simulations, the current experiments sufficiently demonstrate/support the main points of the paper. While I did not check proofs/appendix in detail, the technical portions of the main paper are, to the best of my knowledge, sound and correct.

Significance: The topic of learning generative models is timely and significant. The proposed method is also a significant contribution in my opinion, more than enough to meet the bar for ICLR.

**Weaknesses:**

There are minor points and questions which I bring up below:

- When the authors mention the difficulty of learning multimodal and heavy-tailed targets on page 1, Hagemann and Neumayer (2021) and Salmona et al (2022) are cited. However, there are other highly relevant literature that should have been cited. These include:

- Concentration of Measure for Distributions Generated via Diffusion Models. R Ghane, A Bao, D Akhtiamov, B Hassibi
- On the Statistical Capacity of Deep Generative Models. E Tam, D Dunson
- Copula & Marginal Flows: Disentangling the Marginal from its Joint. M Wiese, R Knobloch, R Korn

- Runtime/computational costs: does the one dimensional approach that the authors propose lead to higher runtime or computational complexity in practice compared to other FM/consistency based approaches? (I am NOT looking for any computational complexity bounds/results, I am mainly interested in just a couple of sentences that comment on the runtime/computational aspects of things so readers can get a rough idea).

**Questions:**

See above section

---

> ### Author Response · Authors · 2025-11-19
>
> Many thanks for the positive assessment! We appreciate the reviewer's recognition of the originality and significance of our work. We are very grateful for the many references to further relevant literature concerning especially heavy-tailed targets, which we now included.
>
> Our point-by-point responses follow below.
>
> >When the authors mention the difficulty of learning multimodal and heavy-tailed targets on page 1, Hagemann and Neumayer (2021) and Salmona et al (2022) are cited. However, there are other highly relevant literature that should have been cited. These include:
>
> >Concentration of Measure for Distributions Generated via Diffusion Models. R Ghane, A Bao, D Akhtiamov, B Hassibi
>
> >On the Statistical Capacity of Deep Generative Models. E Tam, D Dunson
>
> >Copula & Marginal Flows: Disentangling the Marginal from its Joint. M Wiese, R Knobloch, R Korn
>
> Thanks a lot for pointing out these highly relevant references. We have expanded our related work discussion in the introduction to include the three suggested papers, which further strengthens the motivation of our paper.
>
> >Runtime/computational costs: does the one dimensional approach that the authors propose lead to higher runtime or computational complexity in practice compared to other FM/consistency based approaches?
>
> We have added a detailed discussion of computational efficiency in the appendix. On CIFAR-10 (dimension $d=3072$) the key findings are:
>
> **Computational cost of the quantile function:**
> - Our quantile architecture requires only $\mathcal{O}(d \cdot K)$ parameters where $K$ is the number of bins in our spline parametrization, which is negligible compared to the UNet velocity network (typically millions of parameters).
> - During joint training (first 55,000 iterations: 50,000 normal training + 5,000 with learning rate decay), our method introduces approximately 3.2% computational overhead.
> - After freezing the quantile (which we do after 55k steps), the overhead drops to approximately 1.2%.
> - Over the full 400k iteration training schedule, the total runtime slowdown is only approximately 1.5%  compared to standard Gaussian baseline with minibatch OT.
> - The componentwise structure enables fully parallel evaluation with $\mathcal{O}(1)$ wall-clock time.
> - All measurements were conducted on an NVIDIA GeForce RTX 4090.
>
> **Effect of the quantile function on training efficiency:**
> - We added an experiment on capacity constraints (reduced network sizes) on MNIST, showing that our learned latent enables smaller networks to maintain good performance while Gaussian baseline degrades significantly.
>
>
> We would like to also note especially with respect to sampling from a heavy tailed target distribution, we were able to improve our results on the funnel distribution and updated Figure 2.

---

### Author Response · Authors · 2025-11-19

We thank all reviewers for their constructive feedback and valuable suggestions! We have revised the manuscript carefully to address all concerns, and welcome any additional feedback.
Modifications in the revised version are marked in blue.
Here are the main general changes:

1. To enhance clarity and better highlight the significance of our work, we revised the abstract and introduction, and reordered Sections 3 and 4. Section 4 presents our main contribution, while Section 3 establishes our underlying motivation, which is independently noteworthy.
2. We incorporated all proposed references. Many thanks for pointing out!
3. We invested big effort to improve the writing of the numerical part and the experimental results including quantitative comparisons which hopefully underlines the potential of our method.
4. We enhanced the appendix by additional explanations and numerical examples.

---

### Meta-Review · Area_Chair_wtf1 · 2026-01-06

**Summary:**

This paper is about the design principle of flow matching. In standard flow matching, the prior distribution is a standard Gaussian. The Gaussian could be quite different from the data distribution. The authors propose to learn a prior distribution that is closer to the data distribution and claim this can lead to better performance. To make the method tractable, the prior distribution is constrained to be a product measure of 1d distribution along each coordinate. Each 1d distribution is parameterized by its quantile function for computation convenience.

**Reviewer Concerns:**

This paper receives polarized comments. While most reviewers agree the research idea is original, several reviewers raise serious concerns on the experiment study. The paper only considers very simple tasks. Moreover, even on these tasks, the results obtained with the proposed method are not competitive. In addition, the paper lacks comparison study with various baselines and ablation study. The paper needs a significant amount of improvement in experiment section to demonstrate the potential advantages of the proposed method. Otherwise, this will be another non-practical generalization of flow matching.

**Reviewer Scores:**

no

---

### Decision · Program_Chairs · 2026-01-26

Reject